# Divergent features of the coenzyme Q: cytochrome *c* oxidoreductase complex in *Toxoplasma gondii* parasites

**Jenni A. Hayward**[1], **Esther Rajendran**[1], **Soraya M. Zwahlen**[1], **Pierre Faou**[2], **Giel G. van Dooren**[1]*

**1** Research School of Biology, Australian National University, Canberra, Australia, **2** Department of Biochemistry and Genetics, La Trobe Institute for Molecular Science, La Trobe University, Melbourne, Australia

\* giel.vandooren@anu.edu.au

**Data Availability Statement:** All relevant data are within the manuscript and its Supporting Information files with the exception of the mass spectrometry proteomics data. The mass

## Abstract

The mitochondrion is critical for the survival of apicomplexan parasites. Several major anti-parasitic drugs, such as atovaquone and endochin-like quinolones, act through inhibition of the mitochondrial electron transport chain at the coenzyme Q:cytochrome *c* oxidoreductase complex (Complex III). Despite being an important drug target, the protein composition of Complex III of apicomplexan parasites has not been elucidated. Here, we undertake a mass spectrometry-based proteomic analysis of Complex III in the apicomplexan *Toxoplasma gondii*. Along with canonical subunits that are conserved across eukaryotic evolution, we identify several novel or highly divergent Complex III components that are conserved within the apicomplexan lineage. We demonstrate that one such subunit, which we term *Tg*QCR11, is critical for parasite proliferation, mitochondrial oxygen consumption and Complex III activity, and establish that loss of this protein leads to defects in Complex III integrity. We conclude that the protein composition of Complex III in apicomplexans differs from that of the mammalian hosts that these parasites infect.

## Author summary

Apicomplexan parasites cause numerous diseases in humans and animals, including malaria (*Plasmodium* species) and toxoplasmosis (*Toxoplasma gondii*). The coenzyme Q: cytochrome *c* oxidoreductase protein complex (Complex III) performs a central role in the mitochondrial electron transport chain of many eukaryotes. Despite being the target of several major anti-apicomplexan drugs, the protein composition of Complex III in apicomplexans was previously unknown. Our work identifies novel proteins in Complex III of apicomplexans, one of which is critical for complex function and integrity. Our study highlights divergent features of Complex III in apicomplexans, and provides a broader understanding of Complex III evolution in eukaryotes. Our study also provides important insights into what sets this major drug target apart from the equivalent complex in host species.

spectrometry proteomics data have been deposited to the ProteomeXchange consortium via the Pride partner repository with the dataset identified PXD018781 and 10.6019/PXD018781.

**Funding:** This work was supported by a Research School of Biology innovation grant to ER and GGvD, a National Health and Medical Research Council Ideas grant (GNT1182369) to GGvD, and an Australian Government Research Training Program Scholarship to JAH. The funders had no role in study design, data collection and analysis, decision to publish, or preparation of the manuscript.

**Competing interests:** The authors have declared that no competing interests exist.

## Introduction

Apicomplexans are a large phylum of intracellular, protozoan parasites that include the causative agents of malaria (*Plasmodium* species) and toxoplasmosis (*Toxoplasma gondii*). These parasites impose major economic and health burdens on human societies, and, in the absence of effective vaccines [1,2], there is a heavy reliance on drugs to treat disease. The parasite coenzyme Q:cytochrome *c* oxidoreductase complex (Complex III of the mitochondrial electron transport chain, ETC) represents one of the major drug targets in these parasites [3,4]. Numerous inhibitors of Complex III, including atovaquone and endochin-like quinolones, are in clinical use or in preclinical development against apicomplexans [5–7].

The ETC consists of a series of protein complexes that are embedded in the inner mitochondrial membrane. Electrons derived from the oxidation of mitochondrial substrates are donated via the action of dehydrogenases to a mobile electron carrier in the inner membrane called coenzyme Q (CoQ). CoQ exchanges electrons with Complex III at the $Q_o$ and $Q_i$ sites of the cytochrome *b* protein of Complex III, in a process called the Q cycle [8]. At the $Q_o$ site, electrons from reduced CoQ are donated to a heme moiety on cytochrome *b*, from where they are either donated on to cytochrome *c*, a mobile carrier protein in the mitochondrial intermembrane space, or donated back to CoQ at the $Q_i$ site. The transfer of electrons to cytochrome *c* occurs via an iron-sulfur cluster and a heme prosthetic group in the Rieske and cytochrome $c_1$ proteins of Complex III, respectively. Electrons from cytochrome *c* are transported on to the cytochrome *c* oxidase complex (Complex IV), which facilitates electron transfer to the terminal electron acceptor, oxygen [9,10]. Electron transport through Complexes III and IV is coupled to the translocation of protons from the mitochondrial matrix into the intermembrane space, thereby generating a proton motive force across the inner membrane. This proton gradient is used for several important mitochondrial processes, including protein and solute import, and driving the activity of ATP synthase (Complex V) to generate ATP [11,12].

It is becoming increasingly apparent that the ETC of myzozoans–a eukaryotic lineage that includes apicomplexan parasites and their closest free living relatives, two phyla of marine algae called chromerids and dinoflagellates–differs considerably from that of other eukaryotes, including the animal hosts that apicomplexans infect [13–15]. For instance, Complexes IV and V from these organisms contain many subunits that lack homologs outside the myzozoan lineage [16–18]. Uncovering such diversity in a canonical mitochondrial process is of interest from both an evolutionary and a therapeutic standpoint. Evolutionarily, these studies provide insights into the diversification of mitochondria since eukaryote lineages diverged from their common ancestor >1.5 billion years ago [19]. Therapeutically, the discovery of novel proteins in a critical pathway such as the ETC highlights differences between parasites and their mammalian hosts that could open avenues for drug development.

The mechanism of action of Complex III inhibitors has been studied extensively, with most exploiting structural differences between the $Q_o$ and/or $Q_i$ sites of Complex III in parasites and host species to selectively target parasites [5–7]. Despite this, many important features of Complex III in apicomplexan parasites, including its protein composition, have not yet been elucidated. Here, we undertake a proteomic analysis of Complex III in *T. gondii*. Along with canonical subunits that are conserved in host organisms, we identified two highly divergent and two novel, apicomplexan-specific Complex III components. We demonstrate that one of the apicomplexan-specific subunits, which we term *Tg*QCR11, is critical for parasite proliferation and ETC function through maintaining Complex III integrity. We conclude that Complex III of apicomplexans contains highly divergent and novel protein subunits, and differs considerably from the equivalent complex in animal hosts.

## Results

### *T. gondii* Complex III contains both canonical and novel protein subunits

The putative mitochondrial processing peptidase alpha subunit (*Tg*MPPα; www.toxodb.org gene ID TGGT1_202680) was shown previously to localize to the mitochondrion of *T. gondii* [20]. In many other eukaryotes, the MPPα protein functions in both the cleavage of mitochondrial-targeting presequences from mitochondrial matrix proteins as they are imported into the mitochondrion, and as one of the so-called Core proteins of Complex III [21]. To facilitate further characterization of *Tg*MPPα, we introduced a TEV-HA tag into the 3' end of the open reading frame of the native *Tg*MPPα locus in *T. gondii* parasites (S1A and S1B Fig). To determine whether *Tg*MPPα exists as part of a protein complex, we extracted proteins from *Tg*MPPα-TEV-HA parasites using 1% (w/v) digitonin, 1% (v/v) Triton X-100 or 1% (w/v) n-Dodecyl β-D-maltoside (DDM) detergents, and separated solubilized protein complexes by blue native (BN)-PAGE. Western blotting with anti-HA antibodies revealed that the primary *Tg*MPPα-TEV-HA complex is ~675 kDa in mass when solubilized in any of the three detergents (Fig 1A), though a fainter secondary complex at ~220 kDa was also observed. By contrast, the molecular mass of *Tg*MPPα-TEV-HA when separated by SDS-PAGE was ~60 kDa (Fig 1B). We conclude that *Tg*MPPα is a component of a ~675 kDa protein complex.

To identify the proteins that comprise the *Tg*MPPα-containing complex, we immunoprecipitated *Tg*MPPα-TEV-HA and associated proteins, then subjected samples to mass spectrometry-based proteomic analyses. As a control for these experiments, we purified the unrelated cytochrome *c* oxidase complex (Complex IV of the ETC). To purify Complex IV, we introduced a TEV-HA tag into the 3' end of the cytochrome *c* oxidase subunit 2a (*Tg*Cox2a; TGGT1_226590) open reading frame (S1C and S1D Fig), immunoprecipitated *Tg*Cox2a-TEV-HA and associated proteins, and performed separate mass spectrometry-based proteomic analyses. Using this approach, we identified five proteins–including *Tg*MPPα–that were enriched in the *Tg*MPPα-TEV-HA immunoprecipitation compared to the *Tg*Cox2a-TEV-HA immunoprecipitation across three independent experiments (Fig 1C and 1D and S1 Table). All five proteins are annotated as canonical components of Complex III, including the Core/ mitochondrial processing peptidase protein *Tg*MPPβ (TGGT1_236210), the iron-sulfur cluster protein *Tg*Rieske (TGGT1_320220), the so-called '14 kDa' protein *Tg*QCR7 (TGGT1_288750), and the cytochrome $c_1$ heme protein *Tg*CytC1 (TGGT1_246540) (S2 Fig). Two other canonical Complex III proteins–the 'hinge' protein *Tg*QCR6 (TGGT1_320140) and the cytochrome *b* protein (*Tg*CytB; TGGT1_362110)–were highly enriched in the *Tg*MPPα immunoprecipitation but excluded from the quantitative analysis in Fig 1C because they were absent from at least one replicate of the *Tg*Cox2a control (Fig 1D and S1 Table). Note that the gene ID for *Tg*CytB (TGGT1_362110) encodes a truncated protein, and may represent one of several fragmented *Tg*CytB pseudogenes encoded by genetic material that occurs as 'junk' DNA in the nuclear genome of *T. gondii*, rather than the actual mitochondrial genome-encoded *Tg*CytB gene [22,23].

Given that two canonical Complex III proteins were excluded from the initial analysis, we reasoned that a more comprehensive assessment of the data might reveal additional subunits that were likewise excluded. We shortlisted six additional proteins that were highly enriched in all three *Tg*MPPα replicates relative to *Tg*Cox2a (either absent from all *Tg*Cox2a samples or > 100-fold more abundant in the *Tg*MPPα purification than in the *Tg*Cox2a purification) (Fig 1D and S1 Table). Of these six proteins, five were annotated as hypothetical proteins (TGGT1_201880, TGGT1_227910, TGGT1_207170, TGGT1_214250, and TGGT1_242780) and one was annotated as a GAF domain-containing protein (TGGT1_270800). We utilized Localization of Organelle Proteins by Isotope Tagging ("LOPIT") cellular localization data

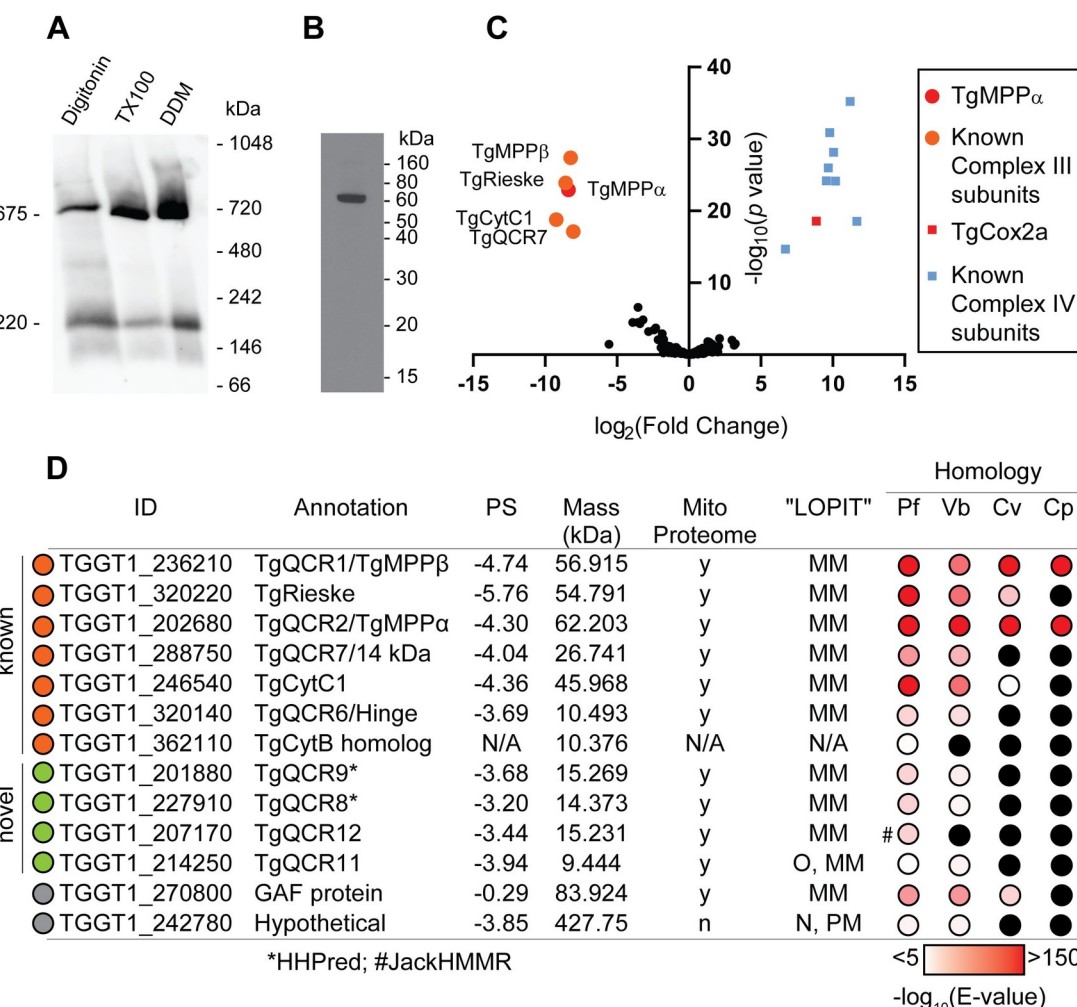

**Fig 1. *Tg*MPPα is part of a ~675 kDa protein complex and co-purifies with known components of Complex III. (A)** Western blot of proteins extracted from *Tg*MPPα-TEV-HA parasites in 1% (w/v) digitonin, 1% (v/v) TX100 or 1% (w/v) DDM-containing lysis buffer, separated by BN-PAGE, and detected with anti-HA antibodies. **(B)** Western blot of proteins extracted from *Tg*MPPα-TEV-HA parasites, separated by SDS-PAGE, and detected with anti-HA antibodies. **(C)** Volcano plot showing the log$_2$ fold change vs -log$_{10}$ *p* values of proteins purified from *Tg*MPPα-TEV-HA vs *Tg*Cox2a-TEV-HA parasites using anti-HA immunoprecipitations and detected by mass spectrometry. To enable statistical comparisons, only proteins detected in all three independent experiments for both parasite lines are depicted. Proteins enriched in the *Tg*MPPα-TEV-HA samples ($p < 0.05$; log$_2$ fold change $> 5$) are labelled and shown in orange circles, with *Tg*MPPα shown in red. Similarly, proteins previously identified as *T. gondii* Complex IV subunits [17] are shown in blue squares, with *Tg*Cox2a shown in red. **(D)** Table summarising the characteristics of proteins identified in proteomic analysis of the *Tg*MPPα-TEV-HA complex (modelled after [18]). Proteins with homology to Complex III proteins are shown in orange circles, novel or divergent Complex III proteins are depicted by green circles, and proteins identified in the initial proteomic analysis but excluded in subsequent analyses are depicted in gray circles. Protein IDs were obtained from ToxoDB and proposed annotations are listed. The phenotype score (PS) of each gene predicts its importance for parasite proliferation, with scores $< $ -2 typically found in genes that are important for proliferation [26]. Detection in the mitochondrial proteome (Mito Proteome, [17]) and its predicted cellular localisation ("LOPIT" [15]) are indicated (y = yes, n = no, N/A = not available, MM = mitochondrial membranes, O = outlier, N = nucleus, PM = plasma membrane). Homology indicates the tBLASTn expected value (E-value) between each *T. gondii* protein sequence and its closest match in *Plasmodium falciparum* (Pf), *Vitrella brassicaformis* (Vb), *Chromera velia* (Cv) or *Cryptosporidium parvum* (Cp) using EuPathDB searches. Black circles indicate a close match could not be identified. *, homology detected using HHPRED; #, homology detected using iterative JackHMMER searches.

[15] coupled with our previously published mitochondrial proteome [17] to assess whether these proteins are likely to be mitochondrial. All the proteins were identified in both the mito-chondrial proteome and the mitochondrial membrane fraction of the "LOPIT" dataset, with

the exception of TGGT1_242780, which we excluded from further consideration (Fig 1D, gray).

We employed a phylogenomic approach to further assess the candidate Complex III proteins. The chromerid *Vitrella brassicaformis* contains the full suite of ETC complexes including Complex III, while *Chromera velia* lacks Complex III but retains the other ETC complexes [14]. We reasoned that if a protein has a homolog in *V. brassicaformis* but not in *C. velia*, this would support the hypothesis that it is a Complex III subunit. In addition, the apicomplexan *Cryptosporidium parvum* contains a highly reduced mitochondrion (termed mitosome) that lacks Complex III [24]. We further hypothesized that 'true' Complex III proteins were likely to be absent from *C. parvum*. To this end, we performed tBLASTn searches of EuPathDB using the identified proteins as queries to identify homologs in *Plasmodium falciparum*, *V. brassicaformis*, *C. velia* and *C. parvum* (Fig 1D). Since *Tg*MPPα and *Tg*MPPβ function together as a mitochondrial peptidase protein complex, separate from their function in Complex III, it is unsurprising that these proteins are found in all four species (Fig 1D). In contrast, most other canonical Complex III proteins have homologs in *P. falciparum* and *V. brassicaformis* but not in *C. velia* and *C. parvum*. An interesting exception is the *Tg*Rieske protein, which has a homolog in *C. velia* (Fig 1D). The *Cv*Rieske protein lacks the long N-terminus typical of other Rieske proteins, but retains the iron-sulfur cluster-containing core, implying this protein may have a secondary function in *C. velia*. A potential *Tg*CytC1 homolog was also identified in *C. velia* (Fig 1D); however, this matched only to the N-terminus of *Tg*CytC1, and lacked key resides including those that mediate heme binding. The *Cv*CytC1 is likely not a functional CytC1 protein.

Of the additional proteins that we shortlisted, one has homologs in both chromerids (TGGT1_270800) and was therefore excluded from further analysis, three have homologs in *V. brassicaformis* but not *C. velia* (TGGT1_201880, TGGT1_227910, TGGT1_214250), and one appeared to be restricted to *T. gondii* (TGGT1_207170) (Fig 1D). Further analysis of TGGT1_207170 using iterative JackHMMER homology searches identified a homolog of this protein in *P. falciparum* and numerous other apicomplexans, but not in *C. parvum*, chromerids or dinoflagellates (Fig 1D). These data suggest that TGGT1_207170 may be an apicomplexan-specific protein.

To investigate whether the four shortlisted proteins (Fig 1D, green) have similarity to known Complex III proteins, we queried each protein against the Protein Data Bank (PDB) using HHPRED, a profile hidden Markov model search tool designed to identify homologous proteins with limited sequence similarity [25]. This analysis predicted that TGGT1_227910 has homology to the yeast Complex III protein QCR8 (percent probability 97.59 and E-value 0.00012) and TGGT1_201880 has homology to yeast QCR9 (percent probability 99.78 and E-value 4.9e-19). We therefore termed these two divergent Complex III subunits *Tg*QCR8 and *Tg*QCR9, respectively (Figs 1D and S2 and S3 and S4). The remaining two proteins were not matched to proteins from PDB with any confidence (percent probabilities <40 and E-values >20). In total, our proteomic analysis identified *T. gondii* homologs of nine out of the ten proteins from the well-studied Complex III of budding yeast, with no homolog to the yeast QCR10 protein apparent in *T. gondii* (S2 Fig). In addition, we identified two proteins that were restricted to the myzozoan or apicomplexan lineages. We termed these proteins *Tg*QCR11 (TGGT1_214250) and *Tg*QCR12 (TGGT1_207170), delineating them from the 10 "QCR" proteins from yeast (Figs 1D and S5 and S6). All the candidate Complex III proteins that were tested in a genome-wide CRISPR screen were predicted to be important for growth of the tachyzoite stage of *T. gondii* [26] (Fig 1D).

To begin to characterize the candidate Complex III proteins experimentally, we introduced FLAG epitope tags into the 3' end of the open reading frames of the *Tg*QCR8, *Tg*QCR9,

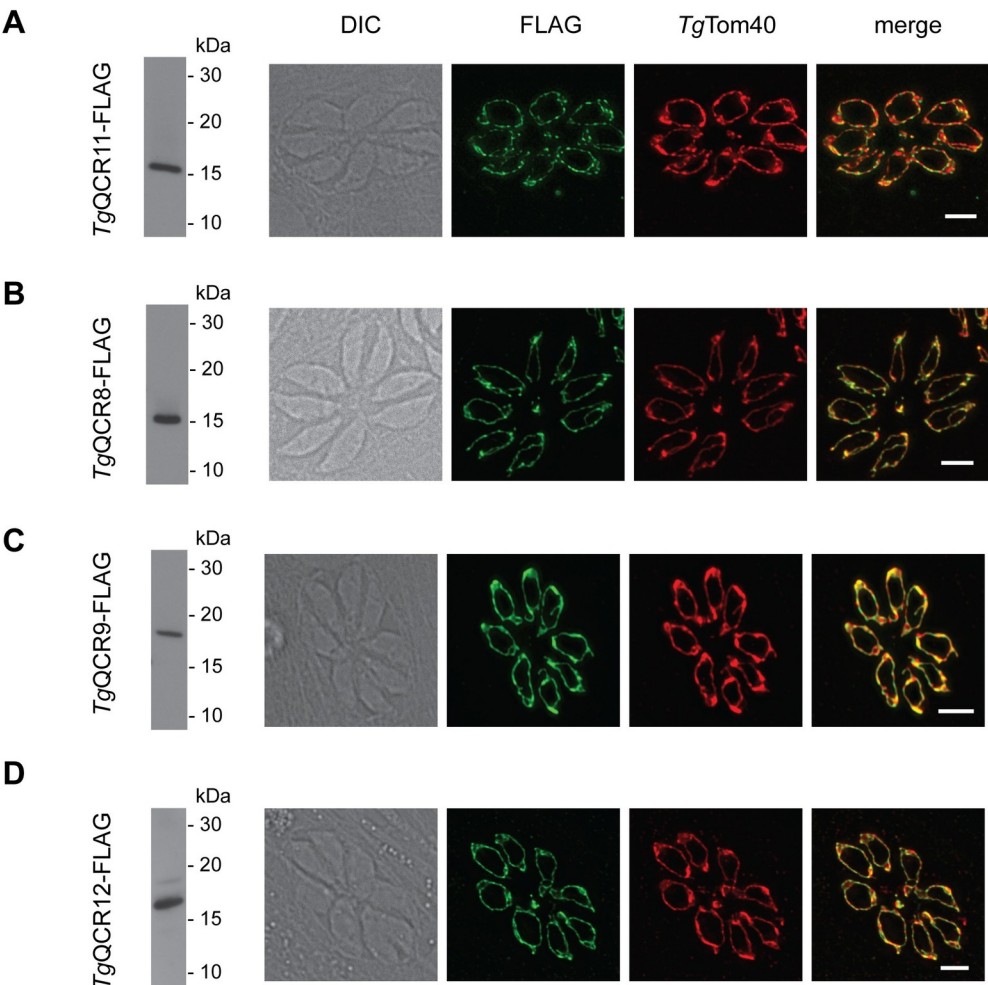

**Fig 2. Candidate Complex III subunits localize to the mitochondrion of *T. gondii*.** (Left) Western blot and (Right) immunofluorescence assay analysis of **(A)** *Tg*MPPα-HA/*Tg*QCR11-FLAG, **(B)** *Tg*MPPα-HA/*Tg*QCR8-FLAG, **(C)** *Tg*MPPα-HA/*Tg*QCR9-FLAG, or **(D)** *Tg*MPPα-HA/*Tg*QCR12-FLAG parasites. Western blots were detected with anti-FLAG antibodies, and immunofluorescence assays were detected with anti-FLAG (green) and the mitochondrial marker anti-*Tg*Tom40 (red) antibodies. Scale bars represent 2 μm.

*Tg*QCR11, and *Tg*QCR12 loci in an existing *Tg*MPPα-HA background strain [20] (S7 Fig). We then undertook western blot analysis and immunofluorescence assays to analyze the expression and cellular localization of these proteins. All four FLAG-tagged proteins were found to be between 15–20 kDa in mass and to localize to the mitochondrion (Fig 2). Interestingly, the relative abundance of these four proteins appeared to differ, with *Tg*QCR11-FLAG and *Tg*QCR8-FLAG the most abundant and *Tg*QCR12-FLAG the least (S8A Fig).

We next sought to determine whether *Tg*QCR11-FLAG, *Tg*QCR8-FLAG, *Tg*QCR9-FLAG and *Tg*QCR12-FLAG exist in protein complexes. To do this, we extracted proteins from the four parasite lines and from the *Tg*MPPα-TEV-HA parasite line using 1% (w/v) DDM, and separated these proteins by BN-PAGE. Western blotting using anti-FLAG antibodies revealed that all four proteins exist in a protein complex of ~675 kDa, approximately the same mass as the *Tg*MPPα-TEV-HA complex when an adjacent lane run on the same gel was probed with anti-HA antibodies (Fig 3A). The relative abundances of these proteins in the ~675 kDa complex reflected their relative abundances in SDS-PAGE western blots (S8B Fig).

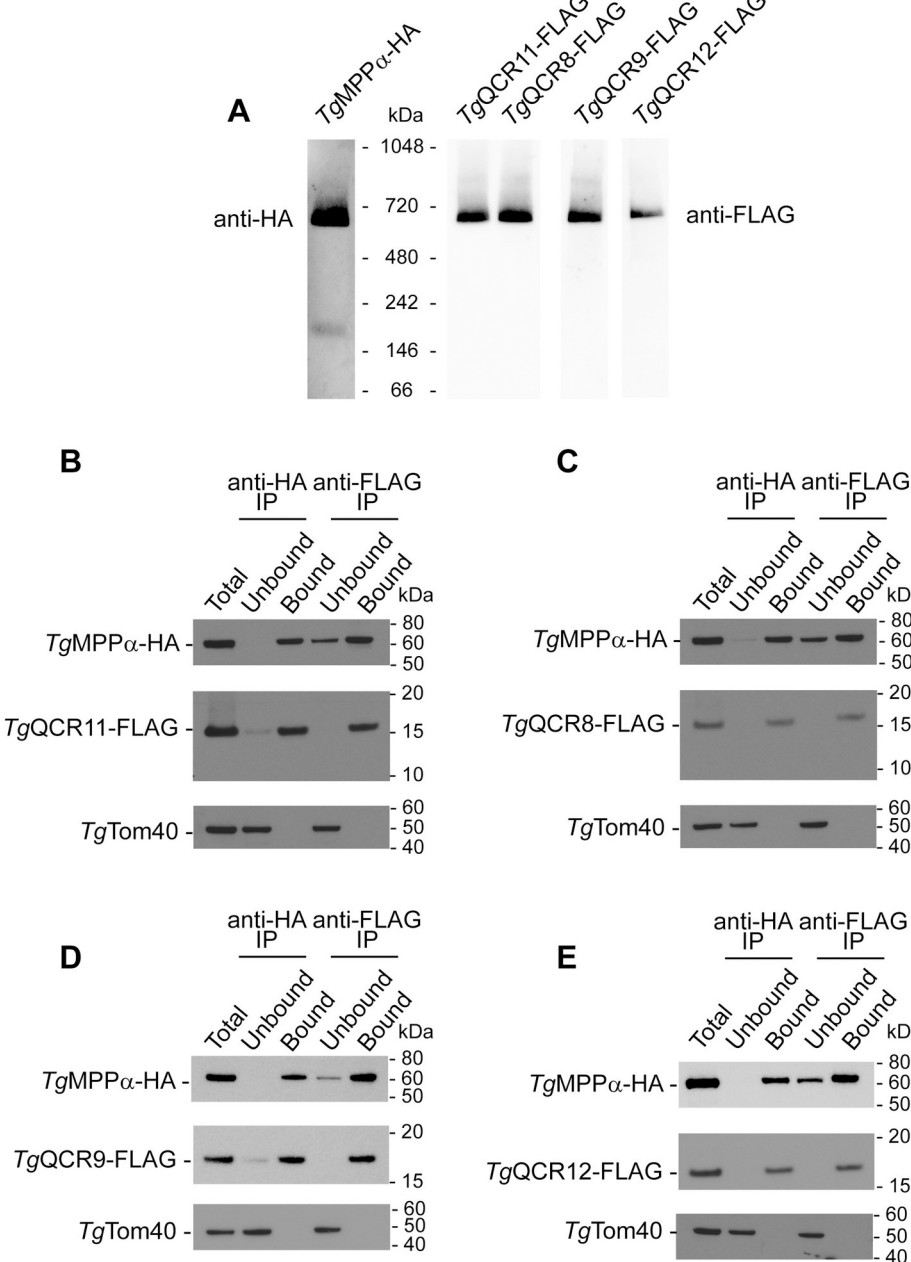

**Fig 3. Candidate Complex III subunits are part of a ~675 kDa protein complex and interact with *Tg*MPPα. (A)** Western blot of proteins extracted from *Tg*MPPα-TEV-HA (left) or *Tg*MPPα-HA/*Tg*QCR11-FLAG, *Tg*MPPα-HA/ *Tg*QCR8-FLAG, *Tg*MPPα-HA/*Tg*QCR9-FLAG and *Tg*MPPα-HA/*Tg*QCR12-FLAG parasites in 1% (w/v) DDM, separated by BN-PAGE, and detected with anti-HA or anti-FLAG antibodies. Images were obtained from a single membrane with lanes cut as indicated and probed with different concentration of antibodies. **(B-E)** Western blots of proteins extracted from **(B)** *Tg*MPPα-HA/*Tg*QCR11-FLAG, **(C)** *Tg*MPPα-HA/*Tg*QCR8-FLAG, **(D)** *Tg*MPPα-HA/ *Tg*QCR9-FLAG or **(E)** *Tg*MPPα-HA/*Tg*QCR12-FLAG parasites, and subjected to immunoprecipitation using anti-HA (anti-HA IP) or anti-FLAG (anti-FLAG IP) antibody-coupled beads. Extracts include samples before immunoprecipitation (Total), samples that did not bind to the anti-HA or anti-FLAG beads (Unbound), and samples that bound to the anti-HA or anti-FLAG beads (Bound). Samples were separated by SDS-PAGE, and probed with anti-HA antibodies to detect *Tg*MPPα-HA, anti-FLAG to detect *Tg*QCRs, and anti-*Tg*Tom40 as a control to detect an unrelated mitochondrial protein.

As a direct test for whether *Tg*QCR11-FLAG, *Tg*QCR8-FLAG, *Tg*QCR9-FLAG and *Tg*QCR12-FLAG proteins are part of the same protein complex as *Tg*MPPα-HA, we performed co-immunoprecipitation experiments. Immunoprecipitation of *Tg*MPPα-HA with anti-HA antibodies co-purified each of the four FLAG-tagged proteins, but not the unrelated mitochondrial protein *Tg*Tom40 (Fig 3B–3E). Likewise, immunoprecipitation of *Tg*QCR11-FLAG (Fig 3B), *Tg*QCR8-FLAG (Fig 3C), *Tg*QCR9-FLAG (Fig 3D) and *Tg*QCR12-FLAG (Fig 3E) with anti-FLAG antibodies co-purified *Tg*MPPα-HA but not *Tg*Tom40. In all instances, we identified a small proportion of *Tg*MPPα-HA in the unbound fraction of the anti-FLAG immunoprecipitations (Fig 3B–3E). This could represent the ~220 kDa complex observed for *Tg*MPPα-HA in BN-PAGE that is absent from the BN-PAGE analyses of the other proteins (Fig 3A). Together, these data indicate that *Tg*MPPα-HA and the four *Tg*QCR proteins exist in the same, ~675 kDa protein complex.

## *Tg*QCR11 is important for *T. gondii* proliferation and mitochondrial oxygen consumption

The existence of *Tg*QCR11-FLAG, *Tg*QCR8-FLAG, *Tg*QCR9-FLAG and *Tg*QCR12-FLAG in a complex with the core protein *Tg*MPPα-HA suggests that these proteins are components of Complex III in *T. gondii*. To elucidate the importance and role of one of the apicomplexan-specific Complex III proteins, we undertook a functional characterization of *Tg*QCR11. We first added a FLAG tag to the 3' end of the *Tg*QCR11 gene (S9A and S9B Fig) and then replaced the native promoter of *Tg*QCR11 with an anhydrotetracycline (ATc)-regulated promoter (S9C and S9D Fig). We termed the resultant ATc-regulated *Tg*QCR11 strain 'r*Tg*QCR11-FLAG'. We also added an HA tag to the 3' end of the *Tg*MPPα gene in this strain (S9E and S9F Fig), resulting in a strain we termed 'r*Tg*QCR11-FLAG/*Tg*MPPα-HA'.

To examine the extent of *Tg*QCR11 knockdown upon the addition of ATc, we cultured r*Tg*QCR11-FLAG/*Tg*MPPα-HA parasites in the absence of ATc or in the presence of ATc for 1–3 days, then separated proteins by SDS-PAGE and performed western blotting. *Tg*QCR11-FLAG expression levels decreased substantially upon the addition of ATc, with only a small amount of protein detectable after 3 days in ATc (Fig 4A). To investigate the impact of *Tg*QCR11-FLAG knockdown on parasite proliferation, we grew wild type (WT) and r*Tg*QCR11-FLAG/*Tg*MPPα-HA parasites in the absence or presence of ATc for 8 days and compared plaque sizes. Plaque sizes in r*Tg*QCR11 but not WT parasites was severely impaired in the presence of ATc (Fig 4B). To determine whether this proliferation defect was specifically due to loss of *Tg*QCR11, we complemented the r*Tg*QCR11-FLAG strain with an additional copy of *Tg*QCR11 expressed from the constitutively expressed α-tubulin promoter. Complementation with constitutively expressed *Tg*QCR11 restored plaque formation in r*Tg*QCR11 parasites cultured in the presence of ATc (Fig 4B), indicating that the proliferation defect we observed upon *Tg*QCR11 knockdown was specifically due to loss of *Tg*QCR11.

Oxygen acts as the final electron acceptor in the ETC. If *Tg*QCR11 is an important component of Complex III, we hypothesized that knockdown of *Tg*QCR11 would lead to defects in oxygen consumption in the parasite. To test this, we utilized a previously established assay to measure oxygen consumption in extracellular parasites using a Seahorse XFe96 extracellular flux analyzer [17]. We grew WT and r*Tg*QCR11-FLAG/*Tg*MPPα-HA parasites in the absence of ATc or in the presence of ATc for 1–3 days then measured their basal mitochondrial oxygen consumption rate (mOCR). While the presence of ATc did not impair basal mOCR in WT parasites, the basal mOCR of r*Tg*QCR11-FLAG parasites was substantially depleted upon *Tg*QCR11 knockdown (Fig 4C), indicating that *Tg*QCR11 is important for mitochondrial oxygen consumption.

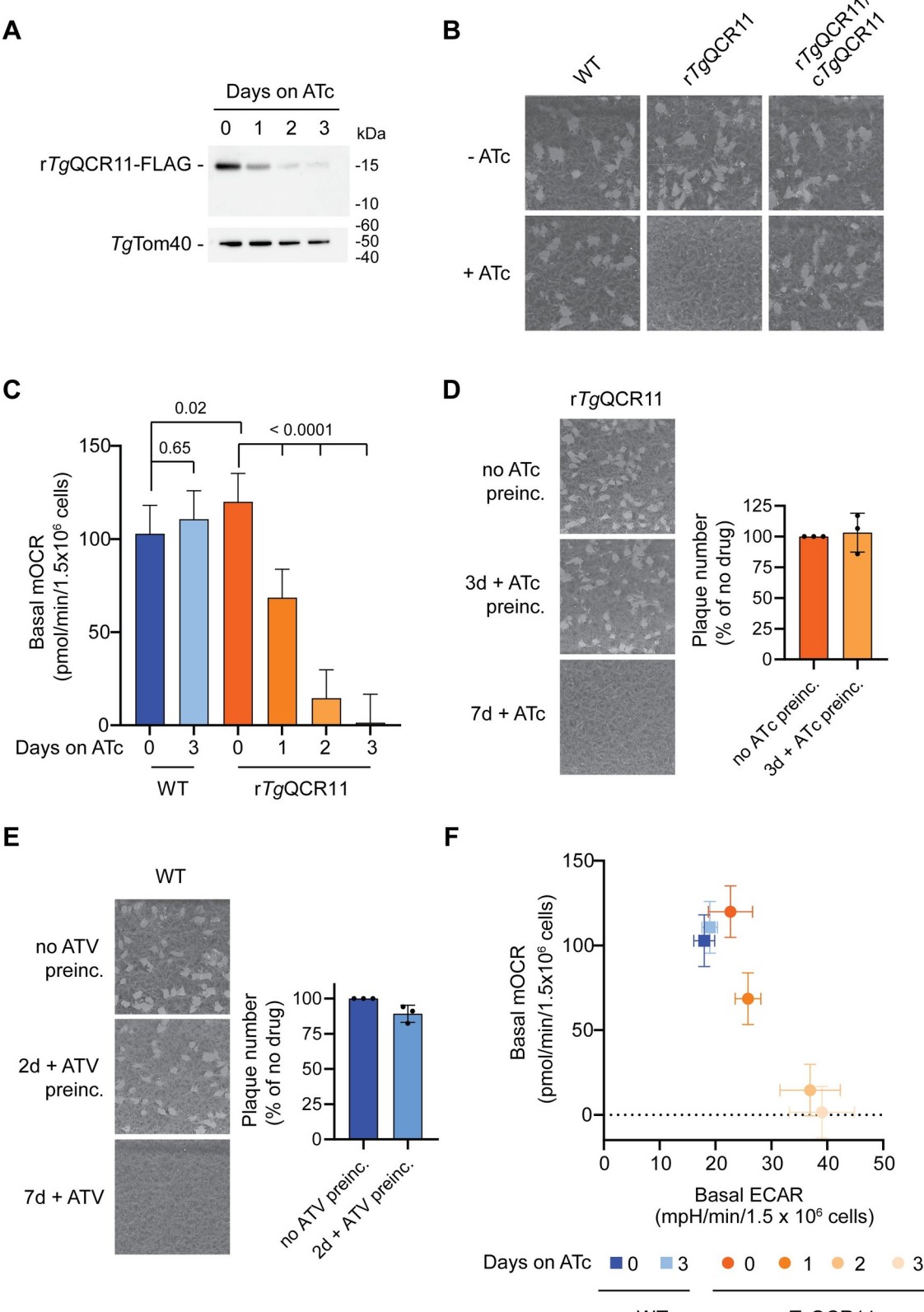

**Fig 4. The apicomplexan-specific Complex III subunit *Tg*QCR11 is important for parasite proliferation and mitochondrial oxygen consumption. (A)** Western blot of proteins extracted from r*Tg*QCR11-FLAG/*Tg*MPPα-HA parasites grown in the absence of ATc, or in the presence of ATc for 1–3 days, separated by SDS-PAGE, and detected using anti-FLAG and anti-*Tg*Tom40 antibodies (loading control). **(B)** Plaque assays measuring growth of WT, r*Tg*QCR11-FLAG/*Tg*MPPα-HA and complemented c*Tg*QCR11-Ty1/ r*Tg*QCR11-FLAG/*Tg*MPPα-HA parasites cultured in the absence (top) or presence (bottom) of ATc for 8 days. Assays are from a single experiment and are representative of 3 independent experiments. **(C)** Basal mitochondrial oxygen consumption rates (mOCR) of WT parasites grown in the absence of ATc or in the presence of ATc for 3 days (blue), and r*Tg*QCR11-FLAG/*Tg*MPPα-HA parasites grown in the absence of ATc or in the presence of ATc for 1–3 days (orange). A linear mixed-effects model was fitted to the data and values depict the least squares mean ± 95% CI of three independent experiments. ANOVA followed by Tukey's multiple pairwise comparisons test was performed, with relevant *p* values shown. **(D)** Plaque assays of r*Tg*QCR11-FLAG parasites grown in the absence of ATc (no ATc preinc; top) for 7 days, pre-incubated in ATc for 3 days before washing out and growing for a further 7 days in the absence of ATc (3d + ATc preinc; middle), or grown in the presence of ATc for all 7 days (7d + ATc; bottom). Quantifications of plaque number as a percent of the no ATc control are depicted to the right of the plaque assays, with bars representing the mean ± standard deviation of 3 independent experiments. **(E)** Plaque assays of WT parasites grown in the absence of atovaquone (no ATV preinc; top), pre-incubated in ATV for 2 days before washing out and growing for 7 days in the absence of ATV (2 days + ATV preinc; middle), or grown in the presence of ATV for all 7 days (7d + ATV; bottom). Quantifications of plaque number as a percent of the no ATV control are depicted to the right of the plaque assays, with bars representing the mean ± standard deviation of 3 independent experiments. **(F)** Basal mOCR versus basal extracellular acidification rate (ECAR) of WT parasites grown in the absence of ATc or in the presence of ATc for 3 days (blue), and r*Tg*QCR11-FLAG/*Tg*MPPα-HA parasites grown in the absence of ATc or in the presence of ATc for 1–3 days (orange). Data depict the mean mOCR and ECAR values ± 95% CI of the linear mixed-effects model (n = 3).

We wondered whether the severe deficiency in mOCR observed upon *Tg*QCR11 knockdown was due to a specific defect in the ETC, or whether it was caused by a more general defect in mitochondrial function or parasite viability. To test whether *Tg*QCR11 knockdown causes any gross defects in mitochondrial morphology, we grew r*Tg*QCR11/*Tg*MPPα-HA parasites in the absence or presence of ATc and performed immunofluorescence assays to visualize both the inner (*Tg*MPPα-HA) and outer (*Tg*Tom40) mitochondrial membranes. This revealed no observable defects in mitochondrial morphology upon *Tg*QCR11 knockdown (S10A Fig).

Next, we tested whether parasites remain viable following knockdown of *Tg*QCR11. To do this, we pre-incubated r*Tg*QCR11 parasites in the presence of ATc for 3 days (a time point when *Tg*QCR11 is knocked down substantially and mOCR is minimal; Fig 4A and 4C), then washed out the ATc and allowed parasites to proliferate in the absence of ATc. After 7 days proliferation, we compared plaque number to parasites that had not been pre-incubated in ATc. Plaque size and number were similar between the ATc-pre-incubated and non-pre-incubated parasites when grown in the absence of ATc, while parasites grown in the presence of ATc for all 7 days underwent minimal proliferation (Fig 4D). These results indicate that *Tg*QCR11 knockdown is reversible and that r*Tg*QCR11 parasites grown in ATc for 3 days have similar viability to parasites grown in the absence of ATc.

We were curious whether treating parasites with the Complex III inhibitor atovaquone might phenocopy *Tg*QCR11 knockdown in terms of parasite viability. To test this, we grew WT parasites in the presence of atovaquone (1 μM, 50 times the IC$_{50}$ reported by [27]) for 2 days, then washed out the drug and grew parasites for a further 7 days (Fig 4E). While parasites grown in the presence of atovaquone for all 7 days did not form visible plaques, parasites that had been pre-incubated in atovaquone for 2 days showed a similar plaque number to the no drug control (Fig 4E). These results mirror those obtained in the genetic experiments and, together, these data imply that parasites remain viable for at least two days without a functional ETC.

To observe what happens to other aspects of parasite metabolism upon *Tg*QCR11 knockdown, we measured the extracellular acidification rate (ECAR) of WT and r*Tg*QCR11 parasites grown in the absence or presence of ATc. We have previously used ECAR as a general indication of parasite metabolic activity [17,28]. ECAR levels of WT and r*Tg*QCR11 parasites grown

in the absence of ATc or WT parasites grown in the presence of ATc were not significantly different (Figs 4F and S10B). By contrast, growth of r*Tg*QCR11 parasites in the presence of ATc for 2 days resulted in a significant *increase* in ECAR (Figs 4F and S10B), indicating that parasites remained metabolically active upon *Tg*QCR11 knockdown. The small but significant increase in ECAR upon *Tg*QCR11 knockdown may indicate that the parasite compensates for loss of ETC activity by upregulating other aspects of cellular metabolism (e.g. glycolysis), though this needs to be studied further.

Together, these data indicate that the defects observed in mitochondrial oxygen consumption upon *Tg*QCR11 knockdown were not due to general defects in mitochondrial morphology, parasite viability or cellular metabolism. We therefore conclude that *Tg*QCR11 has an important and specific role in the ETC of *T. gondii* parasites, consistent with its association with Complex III.

## *Tg*QCR11 is important for the function and integrity of Complex III

To establish whether *Tg*QCR11 is important for Complex III function in *T. gondii*, we sought to undertake a more direct interrogation of the functionality of various ETC components upon *Tg*QCR11 knockdown. To do this, we established an XFe96 flux analyzer-based assay that enabled us to measure substrate-dependent mOCR using digitonin-permeabilized extracellular parasites. Permeabilizing the plasma membrane of parasites with digitonin allows the passage of substrates with polar functional groups into parasites, where they can feed electrons into the ETC either directly or via mitochondrial metabolism. Supplying different combinations of substrates and inhibitors during the assay enables an assessment of the functionality of different ETC complexes (Fig 5A; [29]).

We grew WT, r*Tg*QCR11 and r*Tg*ApiCox25 (an ATc-regulatable strain that enables knockdown of the Complex IV protein *Tg*ApiCox25; [17]) parasites in the absence of ATc or in the presence of ATc for 1–3 days. Parasites were starved for 1 hour in base medium to deplete endogenous substrates, and then permeabilized with 0.002% (w/v) digitonin. The five readings taken before injection of substrate show that permeabilized parasites have very low OCR (Fig 5B), indicating that the 1 hour starvation successfully depleted endogenous substrates. Injection of the tricarboxylic acid (TCA) cycle substrates malate and glutamate caused an almost instantaneous increase in OCR in WT, r*Tg*QCR11 and r*Tg*ApiCox25 parasites that were cultured in the absence of ATc (Fig 5B). As this OCR could be abolished by subsequent injection of the Complex III inhibitors antimycin A and atovaquone (Fig 5B), these data indicate that the ETC is functional in these parasites. The malate/glutamate-elicited mOCR of WT parasites grown for 3 days in the presence of ATc was not significantly different to WT parasites grown in the absence of ATc (Fig 5C), indicating that ATc itself does not impact ETC function. By contrast, knockdown of *Tg*QCR11 and *Tg*ApiCox25 caused significant decreases in malate/glutamate-dependent mOCR (Fig 5B and 5C).

We next asked whether defects in malate/glutamate-dependent mOCR upon *Tg*QCR11 or *Tg*ApiCox25 knockdown occurred upstream or downstream of cytochrome *c*. To test this, we injected the cytochrome *c* substrate N,N,N′,N′-tetramethyl-p-phenylenediamine dihydrochloride (TMPD). Reduced TMPD donates electrons directly to cytochrome *c*, downstream of Complex III, and should therefore rescue mOCR in parasites with an ETC defect upstream of cytochrome *c* (Fig 5A). By contrast, a Complex IV defect should not be rescued by TMPD since Complex IV is downstream of cytochrome *c*. Injection of TMPD caused an increase in mOCR in r*Tg*QCR11 parasites cultured in the presence of ATc (Fig 5B). By contrast, r*Tg*ApiCox25 parasites cultured for 2 or 3 days on ATc had very little TMPD-dependent mOCR (Fig 5B). Calculating the fold stimulation of mOCR by TMPD relative to malate/

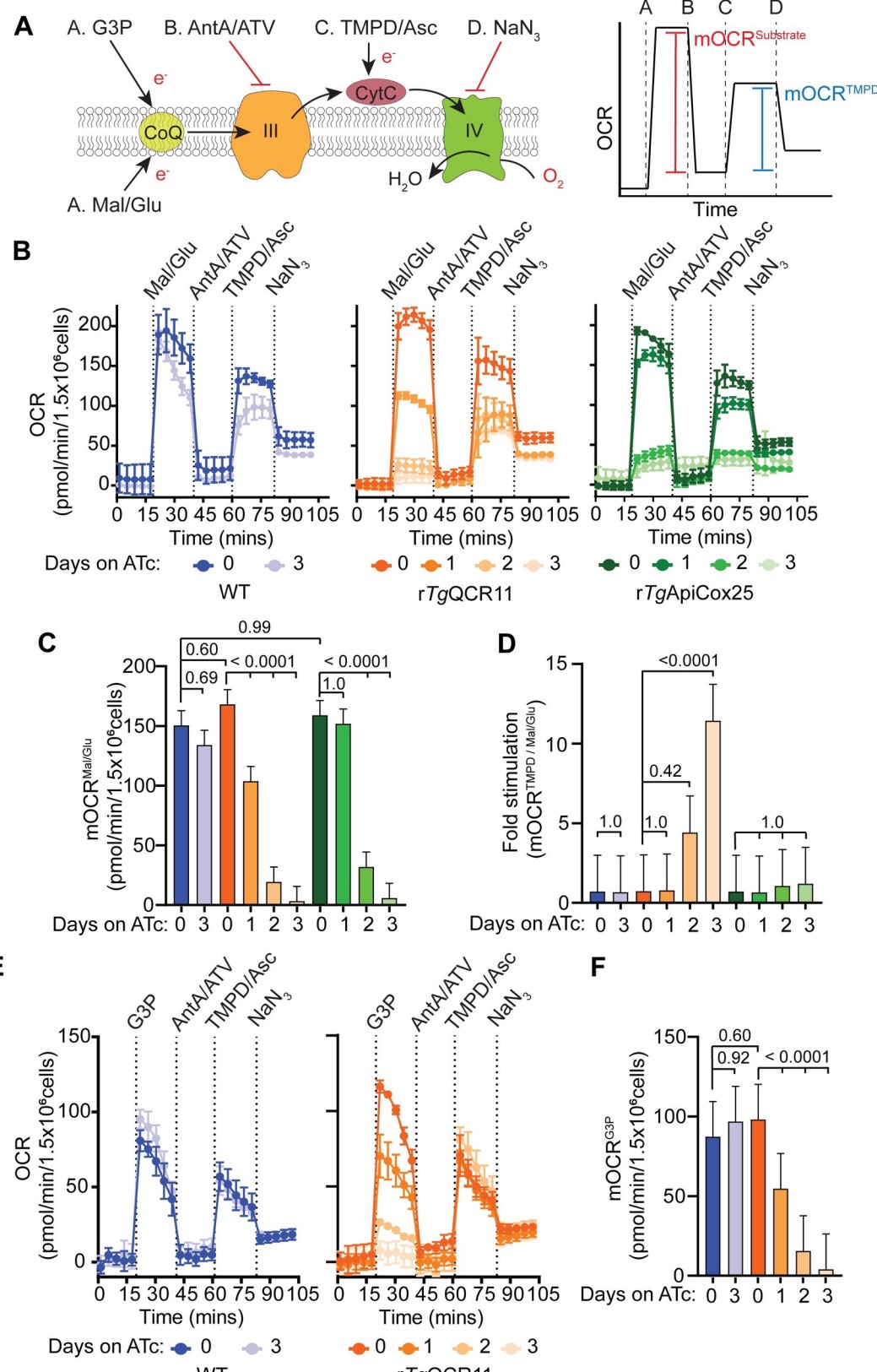

**Fig 5. Loss of *Tg*QCR11 leads to a specific defect in Complex III function. (A)** Schematic diagram of the assay measuring OCR in digitonin-permeabilized parasites, with inset (right) depicting a mock oxygen consumption rate (OCR) versus time graph to illustrate the typical response of WT parasites. Parasites are starved for 1 hour in base media to deplete endogenous energy sources, then permeabilized with 0.002% (w/v) digitonin before being subjected to the following injections of substrates or inhibitors: Port A, malate and glutamate (Mal/Glu) or glycerol 3-phosphate (G3P); Port B, antimycin A and atovaquone (AntA/ATV); Port C, TMPD and ascorbate (TMPD/Asc); Port D, sodium azide (NaN$_3$). The mitochondrial OCR (mOCR) elicited by a substrate (red line, mOCR$^{substrate}$) and the mOCR elicited by TMPD/Asc (blue line, mOCR$^{TMPD}$) are then calculated from these data. CytC, cytochrome *c*; CoQ, coenzyme Q; III, Complex III; IV, Complex IV; e$^-$, electrons. **(B)** Representative traces depicting OCR over time when supplying Mal/Glu (10 mM) as an energy source. WT (blue), r*Tg*QCR11-FLAG/*Tg*MPPα-HA (orange) and r*Tg*ApiCox25-HA (green) parasites were grown in the absence of ATc or in the presence of ATc for 1–3 days. Data represent the mean ± SD of three technical replicates, and are representative of three independent experiments. **(C)** Mal/Glu elicited mOCR (mOCR$^{Mal/Glu}$) of WT (blue), r*Tg*QCR11-FLAG/*Tg*MPPα-HA (orange) and r*Tg*ApiCox25-HA (green) parasites that were grown in the absence of ATc or in the presence of ATc for 1–3 days. A linear mixed-effects model was fitted to the data and values depict the least squares mean ± 95% CI from three independent experiments. ANOVA followed by Tukey's multiple pairwise comparisons test was performed, with relevant *p* values shown. **(D)** Fold stimulation of mOCR by TMPD relative to Mal/Glu in WT (blue), r*Tg*QCR11-FLAG/*Tg*MPPα-HA (orange) and r*Tg*ApiCox25-HA (green) parasites that had been grown in the absence of ATc or in the presence of ATc for 1–3 days (mean ± 95% CI of the linear mixed-effects model; n = 3). ANOVA followed by Tukey's multiple pairwise comparisons test was performed, with relevant *p* values shown. **(E)** Representative traces depicting OCR over time when supplying the TCA cycle-independent substrate G3P (10 mM) as an energy source. WT (blue) and r*Tg*QCR11-FLAG/*Tg*MPPα-HA (orange) parasites were grown in the absence of ATc or in the presence of ATc for 1–3 days. Data represent the mean ± SD of three technical replicates, and are representative of three independent experiments. **(F)** G3P elicited mOCR (mOCR$^{G3P}$) of WT (blue) and r*Tg*QCR11-FLAG/*Tg*MPPα-HA (orange) parasites that were grown in the absence of ATc or in the presence of ATc for 1–3 days (mean ± 95% CI of the linear mixed-effects model; n = 3). ANOVA followed by Tukey's multiple pairwise comparisons test was performed, with relevant *p* values shown.

glutamate-dependent mOCR indicates that r*Tg*QCR11 parasites grown for 3 days on ATc have 10-fold greater stimulation of mOCR by TMPD relative to malate (Fig 5D). This demonstrates that Complex IV activity is largely retained upon *Tg*QCR11 knockdown and implies that the defect in ETC activity upon loss of this protein occurs upstream of cytochrome *c*. By contrast, r*Tg*ApiCox25 parasites grown for 3 days on ATc have little stimulation of mOCR by TMPD relative to malate/glutamate, indicating that loss of *Tg*ApiCox25 leads to a defect in Complex IV activity (Fig 5D).

The preceding data indicate that loss of *Tg*QCR11 leads to defects in the ETC upstream of cytochrome *c*. Since malate/glutamate can feed into the ETC via the TCA cycle [13,30], it is conceivable that defects we observe in mOCR upon *Tg*QCR11 knockdown could be the result of defects in the TCA cycle rather than a selective defect in Complex III. To address this, we measured mOCR using the substrate glycerol 3-phosphate, which donates electrons to CoQ independently of the TCA cycle (Fig 5A; [13,30]). Knockdown of *Tg*QCR11 caused a similar decrease in glycerol 3-phosphate-dependent mOCR compared to malate/glutamate-dependent mOCR (Fig 5E and 5F). Since mOCR elicited by two different substrates, one of which is independent of the TCA cycle, was impaired by knockdown of *Tg*QCR11, we conclude that loss of *Tg*QCR11 likely leads to a selective defect in Complex III activity.

Given the severe defect in mOCR observed upon *Tg*QCR11 knockdown, we wondered whether the integrity of Complex III is compromised by loss of *Tg*QCR11. To assess this, we set out to measure the effects of *Tg*QCR11 knockdown on several Complex III proteins, including *Tg*MPPα, the apicomplexan specific subunit *Tg*QCR12, the supernumerary subunit *Tg*QCR8, and the catalytic subunit *Tg*CytC1. We integrated TEV-HA tags into the 3' ends of the *Tg*QCR8 and *Tg*QCR12 open reading frames (S11 Fig), or an HA tag into the 3' end of the *Tg*CytC1 open reading frame in the r*Tg*QCR11-FLAG parasite line. We performed co-immunoprecipitation analysis and verified that *Tg*CytC1 and *Tg*QCR11 are part of the same protein complex (S12 Fig). We then grew r*Tg*QCR11-FLAG/*Tg*MPPα-HA, r*Tg*QCR11-FLAG/*Tg*QCR8-TEV-HA, r*Tg*QCR11-FLAG/*Tg*QCR12-TEV-HA and r*Tg*QCR11-FLAG/*Tg*CytC1-HA parasites in the absence of ATc or in the presence of ATc for 1–3 days and

assessed Complex III integrity by BN-PAGE. Strikingly, the ~675 kDa complex of all four proteins was depleted upon *Tg*QCR11 knockdown (Figs 6A–6D and S13), suggesting that Complex III assembly and/or stability may be impaired upon loss of *Tg*QCR11. We wondered whether the abundance of Complex III proteins may also be affected by *Tg*QCR11 knockdown, and assessed this by SDS-PAGE and western blotting. Interestingly, while the abundance of *Tg*MPPα remained consistent (Figs 6E and S13), the abundance of *Tg*QCR12, *Tg*QCR8 and *Tg*CytC1 decreased by day 2–3 on ATc (Figs 6F–6H and S13), suggesting that knockdown of *Tg*QCR11 decreases the abundance of some Complex III proteins but not others. It is possible that *Tg*MPPα abundance remains unchanged because, in addition to being a component of Complex III, it exists in a lower mass, ~220 kDa complex that our BN-PAGE analysis indicated is retained upon *Tg*QCR11 knockdown (Fig 6A). This ~220 kDa complex could represent the mitochondrial processing peptidase in these parasites. Together, these data indicate that *Tg*QCR11 is a novel, apicomplexan-specific subunit of Complex III in *T. gondii* that is critical for Complex III function by maintaining the integrity of this protein complex.

## Discussion

In this study, we characterized Complex III of the mitochondrial ETC in *T. gondii* parasites, which we demonstrate exists as a ~675 kDa complex comprising 11 protein subunits (Fig 1). Our data are consistent with an independent, parallel study by MacLean and colleagues, who identified the same 11 subunits in a broader proteomic analysis of mitochondrial respiratory chain complexes in *T. gondii* [31]. The number of subunits we identified in *T. gondii* Complex III is similar to that reported for the bovine [32–34], yeast [35] and plant [21,36] complexes (11, 10 and 10 subunits, respectively), and the mass is similar to the fully-assembled yeast complex (670 kDa [37]). The overall architecture of Complex III in eukaryotes is broadly conserved as a homodimer, with one copy of each subunit per monomer [32–35,38], and this may also be the case for *T. gondii*. The sum of the predicted masses of the 11 *T. gondii* Complex III subunits is ~350 kDa (assuming *Tg*CytB mass to be ~41 kDa; [22]), giving a mass of ~700 kDa for the Complex III homodimer, which is in the ballpark of the ~675 kDa we observed by BN-PAGE. A caveat to this conclusion is that we observed different abundances in some Complex III subunits (S8 Fig), raising the possibility that not all subunits exist in a strict 1:1 stoichiometry per monomer.

In our proteomic analysis of *T. gondii* Complex III, we identified numerous canonical subunits, including the three electron transporting subunits (*Tg*Rieske, *Tg*CytB and *Tg*CytC1), the two core proteins (*Tg*MPPα and *Tg*MPPβ), and two known additional (or so-called 'supernumerary') subunits, the 'hinge' protein *Tg*QCR6 and '14 kDa' protein *Tg*QCR7. Our analysis also identified two highly divergent supernumerary subunits, *Tg*QCR8 and *Tg*QCR9, which were only identifiable using the HHPRED search tool. This indicates that, like several supernumerary proteins in Complexes IV and V of *T. gondii* [16–18], the ~1.5 billion years of evolution since the common ancestor of apicomplexans and other eukaryotes has resulted in considerable divergence in the sequences of these ETC proteins.

Intriguingly, our analysis also identified two novel Complex III subunits (*Tg*QCR11 and *Tg*QCR12) that are restricted to organisms closely related to *T. gondii*. *Tg*QCR11 has homologs in other myzozoans–a lineage which comprises apicomplexans, as well as their closest free-living relatives, the chromerids and dinoflagellates–whereas *Tg*QCR12 homologs are restricted to apicomplexans. These observations fit with an emerging narrative that mitochondria of myzozoans have evolved numerous proteins and functions that are unique amongst eukaryotes [15,17,39,40]. The reasons for these novelties are unclear, but it is conceivable that the evolution of proteins like *Tg*QCR11 and *Tg*QCR12 were necessitated by evolutionary pressures

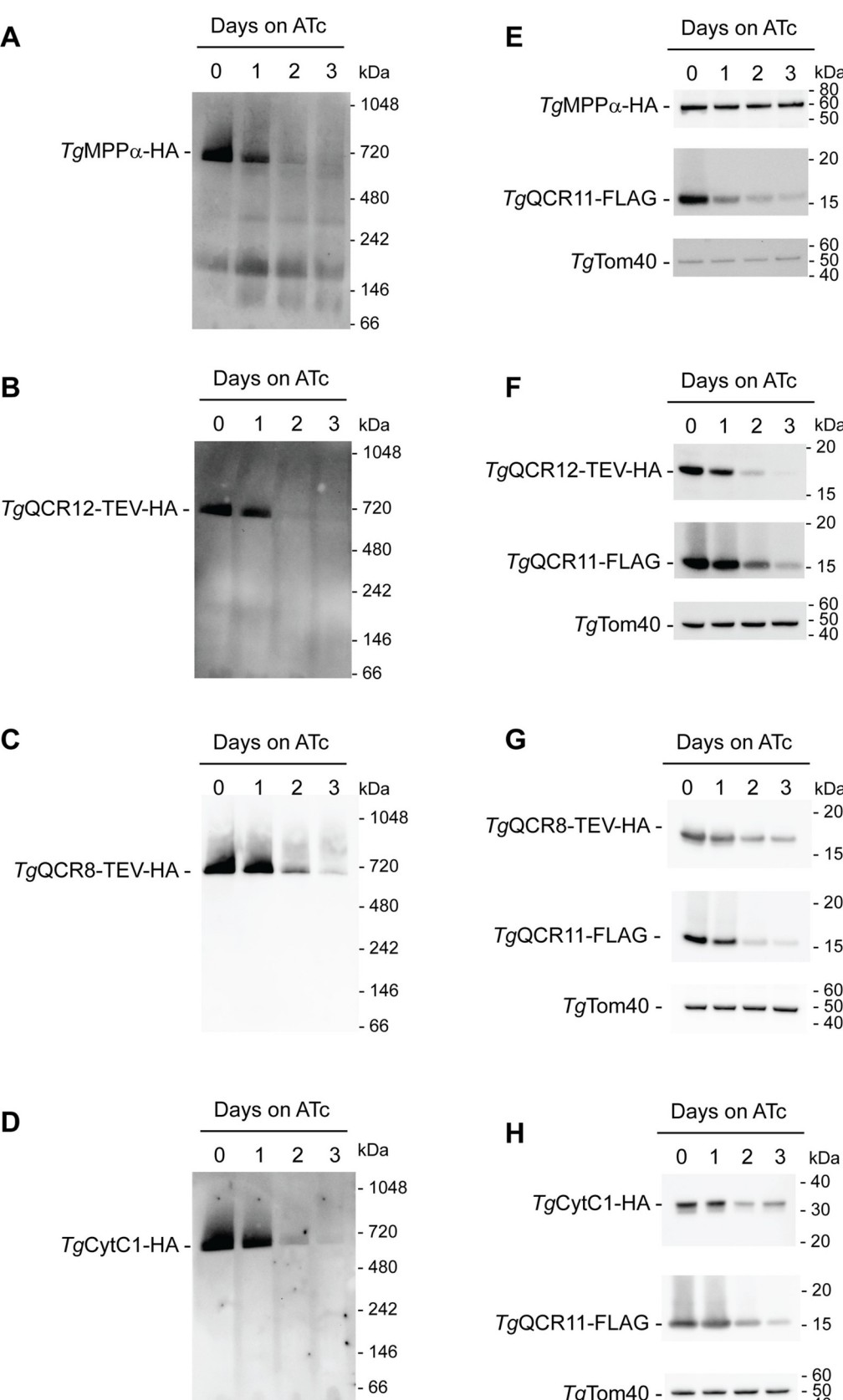

**Fig 6. *Tg*QCR11 is important for Complex III integrity.** (A-D) Western blots of proteins extracted from (A) r*Tg*QCR11-FLAG/*Tg*MPPα-HA, (B) r*Tg*QCR11-FLAG/*Tg*QCR12-TEV-HA, (C) r*Tg*QCR11-FLAG/*Tg*QCR8-TEV-HA and (D) r*Tg*QCR11-FLAG/*Tg*CytC1-HA parasites grown in the absence of ATc or in the presence of ATc for 1–3 days. Samples were prepared in 1% (w/v) DDM, separated by BN-PAGE, and detected with anti-HA antibodies. (E-H) Western blots of proteins extracted from (E) r*Tg*QCR11-FLAG/*Tg*MPPα-HA, (F) r*Tg*QCR11-FLAG/*Tg*QCR12-TEV-HA, (G) r*Tg*QCR11-FLAG/*Tg*QCR8-TEV-HA and (H) r*Tg*QCR11-FLAG/*Tg*CytC1-HA parasites that had been grown in the absence of ATc or in the presence of ATc for 1–3 days. Samples were separated by SDS-PAGE, and probed with anti-HA, anti-FLAG and anti-*Tg*Tom40 (loading control) antibodies. Western blots shown are representative of at least two independent experiments, with matched BN-PAGE and SDS-PAGE samples prepared from the same experiment. Quantifications of protein abundances are depicted in S13 Fig.

for greater ETC efficiency and/or improved mitochondrial energy generation in the marine (and later parasitic) niches in which these organisms evolved. We cannot rule out the possibility that *Tg*QCR11 and *Tg*QCR12 homologs do exist in other lineages of eukaryotes, but have diverged to the extent that they can no longer be detected through even the most sophisticated homology-based searches. In future, obtaining a Complex III structure from apicomplexans will help reconcile these possibilities and shed further light on how the highly diverged and novel subunits identified in this study contribute to Complex III function.

In other eukaryotes such as yeast, the supernumerary subunits of Complex III contribute to complex assembly and/or stability [41,42]. For example, yeast QCR9 (which is homologous to *Tg*QCR9) is important for maintaining the integrity of Complex III [37,43,44], and yeast QCR10 (for which we identified no homolog in *T. gondii*) appears to have a role in stabilising the catalytic Rieske subunit in the complex [45]. Supernumerary subunits have also been proposed to contribute to the assembly of so-called "supercomplexes" between Complex III and other respiratory complexes [46,47]. For example, the yeast Complex III subunits Cor1 (the homolog of *Tg*MPPβ) and QCR8 (the homolog of *Tg*QCR8) interact with Complex IV within these supercomplexes [46,47]. We demonstrate that loss of *Tg*QCR11 leads to severe defects in parasite proliferation and mitochondrial oxygen consumption (Fig 4), decreased abundance of other Complex III subunits, and the disappearance of the ~675 kDa Complex III (Fig 6). These observations are all consistent with *Tg*QCR11 playing a critical role in maintaining Complex III integrity by mediating the assembly and/or the stability of this complex, much like supernumerary Complex III subunits from yeast. The exact role of *Tg*QCR11 in these processes, however, remains elusive. Future functional analyses of *Tg*QCR11, including an examination of its position within the structure of Complex III, may address its actual role in the complex. Much like our findings with *Tg*QCR11, the parallel study by MacLean and colleagues found that knockdown of *Tg*QCR8, *Tg*QCR9 and *Tg*QCR12 led to defects in the assembly of the *Tg*Rieske subunit into the complex [31], suggesting that the various novel and divergent components are all required for integrity of Complex III in *T. gondii*.

In this study, we have developed a powerful suite of assays to probe different stages and complexes of the ETC in *T. gondii* parasites. These assays rely on feeding permeabilized parasites with specific ETC substrates and inhibitors at set times during the assays. We used these assays to demonstrate that knockdown of *Tg*QCR11 caused a specific defect in Complex III function (Fig 5), since 1) the mOCR elicited by two independent substrate combinations (malate/glutamate and glycerol 3-phosphate) was decreased upon *Tg*QCR11 knockdown, and 2) the fold stimulation of TMPD-dependent mOCR compared to malate/glutamate-dependent mOCR was high. This is consistent with *Tg*QCR11 functioning downstream of CoQ yet upstream of cytochrome *c* (i.e. in Complex III). By contrast, the Complex IV protein *Tg*ApiCox25 [17] had a low-fold simulation of TMPD-dependent mOCR compared to malate/glutamate-dependent mOCR, consistent with Complex IV functioning downstream of cytochrome *c* (i.e. in Complex IV). In future, these assays will enable an in-depth characterization of the

function of ETC proteins and complexes, and provide a detailed understanding of the contribution of mitochondrial (and broader parasite) biosynthetic and metabolic pathways to the ETC and energy generation in these parasites. We also note that these approaches lend themselves to drug screening approaches for pin-pointing the target of ETC inhibitors in these parasites.

We observed that parasites remained viable after knockdown of *Tg*QCR11 (Fig 4D), and that this was phenocopied by 2 day treatment with the Complex III inhibitor atovaquone (Fig 4E). These findings suggest that, at least in the short term and at the inhibitor concentration tested, parasites can survive but do not thrive in the absence of mitochondrial respiration. Precisely how parasites achieve this state remains uncertain, though the measurable increases in ECAR observed upon *Tg*QCR11 knockdown (Figs 4F and S10B) suggests that parasites may respond to the impairment of Complex III by upregulating other metabolic pathways (such as glycolysis). Our earlier study of Complex IV demonstrated that parasites remain viable after loss of a key protein in this complex (*Tg*ApiCox25) [17], suggesting that parasite persistence is not specific to impairment of Complex III but is a general phenomenon when targeting the ETC. Determining the mechanism and duration of parasite persistence following the impairment of mitochondrial respiration has potentially important implications for therapeutics that target Complex III. Several clinical studies reported that 12–26% of patients experienced relapse of *T. gondii* infection after 1 year, and up to 75% after 6 years, following atovaquone therapy [48–50]. While the inability of atovaquone to eliminate *T. gondii* infection could be due to many factors, including the conversion of disease-causing tachyzoites to latent bradyzoite-containing tissue cysts [5], it is conceivable that the short-term persistence following ETC impairment may enable parasites to survive initial treatment until bradyzoite differentiation and/or atovaquone resistance arises.

Our work highlights the divergence of mitochondrial ETC Complex III composition in apicomplexan parasites, providing important insights into what sets this major drug target apart from the equivalent complex in host species. Future studies can now build on our findings to reveal how novel subunits of this complex contribute to Complex III function and druggability.

## Materials and methods

### Host cell and parasite culture

*T. gondii* tachyzoites were cultured in human foreskin fibroblasts (HFF), as previously described [28,51]. In paired anhydrotetracycline (ATc) knockdown experiments, ATc (0.5 μg/ml) or ethanol (vehicle control) was added to the media on required days. Plaque assays were performed as described previously [51], with 500 parasites added per flask and incubated for 7–8 days before staining with crystal violet. In the atovaquone washout experiments, 500 parasites were added per flask and allowed to invade host cells for 4 hours before addition of atovaquone (1 μM) to the flask. After 2 days incubation in the presence of atovaquone, the medium was removed, flasks were washed twice with atovaquone-free medium, and parasites allowed to grow for a further 7 days to allow plaques to develop before staining with crystal violet.

### Genetic modifications of *T. gondii*

TATiΔ*ku80* strain parasites [52] were used as the parental cell line in this study. All genetically modified parasites were cloned by flow cytometry before being characterized.

To incorporate a 3' hemagglutinin tag containing a tobacco etch virus protease cleavage site (TEV-HA tag) into the loci of *Tg*MPPα and *Tg*Cox2a, we generated a vector expressing a single guide RNA (sgRNA) targeting the region around the stop codon of *Tg*MPPα and used an

existing sgRNA-expression plasmid to target *Tg*Cox2a [17]. To generate the *Tg*MPPα-targeting vector, we modified the pSAG1::Cas9-U6::sgUPRT plasmid (Addgene plasmid # 54467; [53]) using Q5 mutagenesis (New England Biolabs) as described previously [53]. For site-directed mutagenesis, we used the primers MPPα 3'rep CRISPR fwd and the Universal Reverse primer (S2 Table). We also amplified a TEV-HA tag containing 50 bp of flanking sequence either side of the *Tg*MPPα or *Tg*Cox2a stop codon, using the primers MPPα tag fwd and MPPα tag rvs or Cox2a 3' rep fwd and Cox2a 3' rep rvs, with a TEV-HA tag template synthesized as a gBlock (IDT; S2 Table). The sgRNA expressing vectors, which also expressed GFP-tagged Cas9, were co-transfected into TATiΔ*ku80* strain parasites along with the TEV-HA tags, with transfections performed as described previously [51]. GFP-positive parasites were selected by flow cytometry and cloned 3 days following transfection, then screened for successful integration using the primers MPPα 3' screen fwd and MPPα 3' screen rvs or Cox2a 3' screen fwd and Cox2a 3' screen rvs (S2 Table and S1 Fig).

To introduce 3' FLAG epitope tags into the loci of *Tg*QCR8, *Tg*QCR9, *Tg*QCR11 and *Tg*QCR12, we generated vectors expressing a sgRNA targeting the region around their stop codons. To do this, we modified the pSAG1::Cas9-U6::sgUPRT vector using Q5 mutagenesis with gene specific 3'rep CRISPR fwd primers and the Universal Reverse primer (S2 Table). We also amplified a FLAG tag containing 50 bp of flanking sequence either side of the stop codon of each gene, using gene specific fwd and rvs primers, with FLAG tag template synthesized as a gBlock (IDT; S2 Table). We co-transfected the plasmid and PCR product into *Tg*MPPα-HA strain parasites [20] and also into TATi/Δ*ku80* strain parasites for *Tg*QCR11, selected GFP positive parasites by flow cytometry 3 days post-transfection, then screened for successful integration using gene specific screening fwd and rvs primers (S2 Table and S7E and S7F and S9A and S9B Figs).

To introduce an ATc-regulated promoter into the *Tg*QCR11 locus, we generated a vector expressing a sgRNA targeting the region around the start codon of *Tg*QCR11. To do this, we modified the vector pSAG1::Cas9-U6::sgUPRT using Q5 mutagenesis with the primers QCR11 5' CRISPR fwd and the universal reverse primer (S2 Table). We also PCR amplified the ATc-regulated promoter plus a 'spacer' region consisting of part of the *T. gondii* DHFR open reading frame and 3' UTR using the pPR2-HA3 vector [54] as template and the primers QCR11 pro rep fwd and QCR11 pro rep rvs (S2 Table), which each contain 50 bp of sequence specific for the *Tg*QCR11 locus. We co-transfected the plasmid and the ATc-regulatable promoter into *Tg*QCR11-FLAG strain parasites, selected GFP positive parasites by flow cytometry 3 days post-transfection, then screened for successful integration of the ATc-regulatable promoter using the primers QCR11 5' screen fwd and QCR11 5' screen rvs (S2 Table and S9C and S9D Fig).

To generate a cell line where *Tg*MPPα is HA tagged in the r*Tg*QCR11-FLAG strain, we introduced a 3' HA tag into the locus of *Tg*MPPα using the vector described earlier that expresses a sgRNA targeting the region around the stop codon of *Tg*MPPα. We also amplified a HA tag containing 50 bp of flanking sequence either side of the *Tg*MPPα stop codon, using the primers MPPα tag fwd and MPPα tag rvs, with HA tag template synthesized as a gBlock (IDT; S2 Table). We co-transfected the plasmid and PCR product into r*Tg*QCR11-FLAG strain parasites, selected GFP positive parasites by flow cytometry 3 days post-transfection, then screened for successful integration using the primers MPPα 3' screen fwd and MPPα 3' screen rvs (S2 Table and S9E and S9F Fig).

To generate a vector that constitutively expressed *Tg*QCR11 for complementing the r*Tg*QCR11 mutant, we ordered a gene block encoding the *Tg*QCR11 open reading frame plus a Ty1 tag (IDT; S2 Table) and performed PCR using the primers QCR11 comp fwd and universal Ty1 rvs (S2 Table). We digested the resulting PCR product with *Bgl*II and *Xma*I and

ligated this into the *Bgl*II and *Xma*I sites of the vector pUDTG. The pUDTG vector was generated by Yi Xue (ANU) by digesting the vector pUgCTH3 [55] with *Apa*I and *Hin*dIII, and ligating the excised UPRT flank into the equivalent sites of the vector pDTG (a kind gift from Chris Tonkin, Walter and Eliza Hall Institute). The resulting "*Tg*QCR11 in pUDT-Ty1" vector contains a pyrimethamine-resistance DHFR cassette, a UPRT flanking sequence for integration into the non-essential UPRT locus of *T. gondii*, and fuses a C-terminal Ty1 epitope tag to the complementing *Tg*QCR11 protein. This vector was linearized in the UPRT flanking sequence with *Mfe*I, transfected into r*Tg*QCR11-FLAG/*Tg*MPPα-HA parasites, and selected on pyrimethamine as described [51].

To generate cell lines expressing either *Tg*QCR8-TEV-HA or *Tg*QCR12-TEV-HA in the rTgQCR11-FLAG parasite strain, we introduced 3' TEV-HA tags into the loci of *Tg*QCR8 and *Tg*QCR12. We replicated the strategy for introducing FLAG tags into these loci that is described above, but instead co-transfected the sgRNA-expressing vectors with TEV-HA tags. We amplified TEV-HA tags containing 50 bp of flanking sequence either side of the stop codons using gene specific fwd and rvs primers, with TEV-HA tag template synthesized as a gBlock (IDT; S2 Table). Following transfection, we selected GFP positive parasites by flow cytometry, then screened for successful integration using gene specific screening fwd and rvs primers (S2 Table and S11 Fig).

## SDS-PAGE, BN-PAGE and immunoblotting

Sodium dodecylsulfate (SDS)-polyacrylamide electrophoresis (PAGE), blue native (BN)-PAGE and immunoblotting were performed as described previously [20,56]. SDS-PAGE samples were separated on a 12% (v/v) acrylamide gel and BN-PAGE samples were separated on a 4–16% (v/v) acrylamide gel. Primary antibodies used included mouse anti-FLAG (1:100–1:2,000 dilution; Sigma clone M2), rat anti-HA (1:2000 dilution; Sigma clone 3F10), and rabbit anti-Tom40 (1:2,000 dilution; [20]). The secondary antibodies used were horseradish peroxidase (HRP)-conjugated goat anti-mouse IgG (Abcam, catalog number ab6789), goat anti-rabbit IgG (Abcam, catalog number ab97051) and goat anti-rat IgG (Abcam, catalog number ab97057). For probing for mouse antibodies on immunoprecipitation western blots, HRP-conjugated anti-mouse TrueBlot ULTRA antibodies (Rockland, catalog number 18-18817-33) were used at 1:2,500 dilution. Blots were imaged using X-ray film or using a ChemiDoc MP imaging system (BioRad).

## Immunoprecipitation and mass spectrometry

Immunoprecipitations were performed as described previously [20], except that parasite samples were solubilized in 1% (v/v) Triton X-100. HA-tagged proteins were purified using anti-HA affinity matrix (Sigma; rat anti-HA clone 3F10 antibodies) and FLAG-tagged proteins were purified using anti-FLAG M2 affinity gel (Sigma; mouse anti-FLAG clone M2 antibodies). For mass spectrometry sample preparation, parasite samples were solubilized in 1% (w/v) DDM, and processed as described previously [17]. Briefly, anti-HA affinity matrix bound with HA-tagged protein complexes were frozen at −80˚C for 1 hr, then eluted at room temperature in 0.2 M glycine containing 1% (w/v) DDM (pH 2.3). Samples were neutralized in ammonium bicarbonate, then extracted in chloroform:methanol as described [57]. After extraction, the pellets were dried and stored at −80˚C before mass spectrometry analysis.

Mass spectrometry was conducted as previously described [17]. Briefly, samples were resuspended in 8 M Urea, 50 mM Tris pH 8.3 followed by reduction and alkylation. Solubilized proteins were submitted to trypsin digestion overnight and the resulting peptides purified using the C18 stage tips procedure. Peptides were reconstituted in 0.1% formic acid and 2%

acetonitrile, loaded onto a trap column and washed for 6 min before switching the precolumn in line with the analytical column. The separation of peptides was performed as previously described (11). Data were collected on a Q Exactive HF Orbitrap mass spectrometer (Thermo-Fisher Scientific) in Data Dependent Acquisition mode using m/z 350–1500 as MS scan range at 60 000 resolution, HCD MS/MS spectra were collected for the 10 most intense ions per MS scan at 15 000 resolution with a normalized collision energy of 28% and an isolation window of 1.4 m/z. Dynamic exclusion parameters were set as follows: exclude isotope on, duration 30 s and peptide match preferred. Other instrument parameters for the Orbitrap were MS maximum injection time 30 ms with AGC target $3 \times 106$, MSMS for a maximum injection time of 110 ms with AGT target of $1.1 \times 104$. The raw data were uploaded into Peaks Studio 10.5 (Bioinformatics Solution Inc., Waterloo, Canada) and processed with de novo peptide sequencing and Peaks DB using the ToxoDB (https://toxodb.org/toxo/) database together with common contaminants (cRAP). For peptides identification, the default settings were with precursor-ion and product-ion tolerances set to 10 ppm and 0.02 Da, respectively. Semispecific trypsin digest with a maximum of 1 missed cleavage was employed and peptides were searched with carbamidomethylation of cysteine set as fixed modification. To limit false-positive peptide identification, the false discovery rate (FDR) applied to peptide-spectrum match (PSM) was set to 1%, and at least 1 unique peptide per protein was used.

## Immunofluorescence assays and microscopy

Immunofluorescence assays were performed as described previously [56]. Primary antibodies used were mouse anti-FLAG (1:500 dilution; Sigma clone M2), rat anti-HA (1:500 dilution; Sigma clone 3F10), and rabbit anti-Tom40 (1:2,000 dilution; [20]). Secondary antibodies used were goat anti-mouse Alexa Fluor 488 (1:500 dilution; Thermo Fisher Scientific, catalog number A-11029), goat anti-rat Alexa Fluor 488 (1:500 dilution; Thermo Fisher Scientific, catalog number A-11006), and goat anti-rabbit Alexa Fluor 546 (1:500 dilution; Thermo Fisher Scientific, catalog number A-11035). Images were acquired on a DeltaVision Elite deconvolution microscope (GE Healthcare) fitted with a 100X UPlanSApo oil immersion objective lens (NA 1.40). Images were deconvolved and adjusted for contrast and brightness using SoftWoRx Suite 2.0 software, and subsequently processed using Adobe Illustrator.

## Seahorse XFe96 extracellular flux analysis

Experiments measuring the oxygen consumption rate (OCR) and extracellular acidification rate (ECAR) of intact extracellular parasites were conducted as described previously [17,28]. We also developed experiments to assess the OCR in digitonin-permeabilized parasites utilizing specific ETC substrates. Briefly, parasites were harvested as for the XFe96 assays on intact parasites. Parasites were then washed once in base medium (Agilent Technologies), resuspended in base medium to $1.5 \times 10^7$ cells/mL and starved for 1 hour at 37°C to deplete endogenous ETC substrates. $1.5 \times 10^6$ parasites were added to wells of Cell-Tak-coated Seahorse XFe96 cell culture plates and adhered to the bottom by centrifugation ($800 \times g$, 3 min). Base medium was removed and replaced with 175 μL mitochondrial assay solution (MAS) buffer (220 mM mannitol, 70 mM sucrose, 10 mM $KH_2PO_4$, 5 mM $MgCl_2$, 0.2% w/v fatty acid-free bovine serum albumin, 1 mM EGTA, and 2 mM HEPES-KOH pH 7.4) containing 0.002% (w/v) digitonin. ETC substrates and inhibitors were loaded into Seahorse XFe96 sensor cartridge ports A-D, and injected into wells during the experiment. OCR measurements were obtained every 3 min for five repeats before and after injection of compounds (prepared in MAS buffer; concentrations given are final concentrations following injection). Port A: FCCP (1 μM) plus substrates. The substrates used were malate plus glutamate (10 mM each) or sn-glycerol

3-phosphate bis(cyclohexylammonium) salt (G3P; 25 mM). Port B: antimycin A and atova-quone (10 μM and 1 μM, respectively). Port C: N,N,N′,N′-tetramethyl-p-phenylenediamine dihydrochloride (TMPD; 0.2 mM) mixed with ascorbic acid (3.3 mM). Port D: sodium azide (NaN$_3$; 10 mM). Substrate-elicited mOCR was calculated by subtracting the non-mitochon-drial OCR (the values following antimycin A and atovaquone injection via Port B) from the OCR value obtained after substrate injection (Port A). Likewise, TMPD-elicited mOCR was calculated by subtracting the non-mitochondrial OCR from the OCR value obtained after TMPD injection (Port C). A minimum of 4 background wells were used in each plate, and 3 technical replicates were used for each condition.

## Data analyses

Data from the Seahorse flux analysis were exported from the Seahorse Wave Desktop software (Agilent Technologies). A linear mixed effects model was applied to the data as described pre-viously [17], setting the error between plates (between experiments) and wells (within experi-ments) as random effects, and the mOCR or ECAR values between cell lines and days on drug (ATc) as fixed effects. Analysis of the least square means of the values was performed in the R software environment. Statistical differences between these values were tested through ANOVA (linear mixed effects), with a post hoc Tukey test.

Data from the *Tg*MPPα-TEV-HA and *Tg*Cox2a-TEV-HA mass spectrometry-based proteo-mic experiment were analyzed in the R software environment using the EdgeR package as described previously [17,58]. In the analysis performed to produce the volcano plot, only pro-teins identified in both data sets and each biological replicate were included, while subsequent analyses also considered proteins that were absent from one or more replicates of the *Tg*Cox2a-TEV-HA control.

## Bioinformatic analyses

Amino acid sequences of *T. gondii* Complex III proteins were accessed from ToxoDB (www.toxodb.org). Initial identification of homologs in the apicomplexan parasites *P. falciparum* and *C. parvum*, and the chromerids *V. brassicaformis* and *C. velia*, was performed through Basic Local Alignment Search Tool (tBLASTn) searches of the EuPathDB Transcripts database (www.eupathdb.org., [59]). Where simple BLAST searches did not identify homologs, addi-tional homology searches were performed using the iterative search tool JackHMMER (www.ebi.ac.uk/Tools/hmmer/search/jackhmmer) and the profile hidden Markov model based search tool HHPRED (https://toolkit.tuebingen.mpg.de/tools/hhpred, [25]). To identify homologs of *Tg*QCR8 and *Tg*QCR9 in yeast, humans, *Arabidopsis* and the parasitic dinoflagel-late *Perkinsus marinus*, we performed JackHMMER and NCBI iterative PSI-BLAST [60] searches. We identified dinoflagellate homologs of *Tg*QCR8, *Tg*QCR9 and *Tg*QCR11 in BLAST searches of *Symbiodinium* spp. genomes available at http://reefgenomics.org [61]. Transmembrane domain predictions were performed using TMHMM [62] and TMPred (https://embnet.vital-it.ch/software/TMPRED_form.html). Multiple protein sequence align-ments of QCR8 (S3 Fig), QCR9 (S4 Fig), *Tg*QCR11 (S5 Fig) and *Tg*QCR12 (S6 Fig) were per-formed using Clustal Omega [63], and the graphical output was generated in BoxShade (https://embnet.vital-it.ch/software/BOX_form.html). All accession numbers are included in the supplementary figure legends.

## Supporting information

**S1 Fig. Generating TEV-HA tagged *Tg*MPPα and *Tg*Cox2a strain parasites. (A)** Diagram depicting the 3' replacement strategy to generate TEV-HA-tagged *Tg*MPPα. A sgRNA was

designed to target the *T. gondii* genome near the stop codon of *Tg*MPPα, and cause a double stranded break. A plasmid containing the sgRNA and GFP-tagged Cas9 endonuclease was co-transfected into *T. gondii* parasites with a PCR product encoding a TEV-HA epitope tag flanked by 50 bp of sequence homologous to the regions immediately up- and down-stream of the *Tg*MPPα stop codon. The homologous repair pathway of the parasite mediates integration of the PCR product into the *Tg*MPPα locus. Forward and reverse primers were used to screen parasite clones for successful integration of the TEV-HA tag at the target site, yielding a 274 bp product in the native locus and a 414 bp product in the modified locus. **(B)** PCR screening analysis using genomic DNA extracted from putative *Tg*MPPα-TEV-HA parasites (clones 1–6). Clones 1–4 and 6 yielded PCR products that indicated that these clones had been successfully modified. Genomic DNA extracted from wild type (WT) parasites was used as a control. **(C)** Diagram depicting the 3' replacement strategy to generate TEV-HA-tagged *Tg*Cox2a. A sgRNA was designed to target near the stop codon of *Tg*Cox2a. A plasmid containing the sgRNA and GFP-tagged Cas9 endonuclease was co-transfected into *T. gondii* parasites with a PCR product encoding a TEV-HA epitope tag flanked by 50 bp of sequence homologous to the regions immediately up- and down-stream of the *Tg*Cox2a stop codon. Forward and reverse primers were used to screen parasite clones for successful integration of the TEV-HA tag at the target site, yielding a 260 bp product in the native locus and a 400 bp product in the modified locus. **(D)** PCR screening analysis using genomic DNA extracted from putative *Tg*Cox2a-TEV-HA parasites (clones 1–6). All 6 clones yielded PCR products that indicated that these clones had been successfully modified. Genomic DNA extracted from wild type (WT) parasites was used as a control.
(TIF)

**S2 Fig. Structure and membrane topology of canonical Complex III subunits.** The yeast Complex III dimer structure (PDB: 3CX5) was imported into the EzMol program [64] and subunits were coloured as indicated, with one monomer of each subunit shown as surface display and the other as ribbons. No homolog of yeast QCR10 was detected in *T. gondii* (black), and no homologs of the *T. gondii* proteins *Tg*QCR11 and *Tg*QCR12 (purple) were found in yeast.
(TIF)

**S3 Fig. QCR8 alignment.** Alignment of QCR8 homologs from *Toxoplasma gondii* (*Tg*QCR8; TGME49_214250), *Plasmodium falciparum* (*Pf*QCR8; PF3D7_0306000), *P. berghei* (*Pb*QCR8; PBANKA_0404400), *Theileria equi* (*Te*QCR8; BEWA_031210), *Babesia bovis* (*Bb*QCR8; BBO-V_IV004300), *Vitrella brassicaformis* (*Vb*QCR8; Vbra_14054), *Symbiodinium microadriaticum* (*Sa*QCR8; Smic7304), *Arabidopsis thaliana* (*At*QCR8; NP_196156), *Homo sapiens* (*Hs*QCR8; NP_055217) and *Saccharomyces cerevisiae* (*Sc*QCR8; NP_012369). Dark shading indicates amino acid identity in ≥70% of the sequences, and light shading indicates amino acid similarity in ≥70% of the sequences. The positions of predicted transmembrane domains in *Tg*QCR8 (TMPred prediction) and *Sc*QCR8 (TMHMM prediction) are indicated by red boxes.
(TIF)

**S4 Fig. QCR9 alignment.** Alignment of QCR9 homologs from *T. gondii* (*Tg*QCR9; TGME49_201880), *P. falciparum* (*Pf*QCR9; PF3D7_0622600), *P. berghei* (*Pb*QCR9; PBANKA_1121500), *T. equi* (*Te*QCR9; BEWA_007140), *B. bovis* (*Bb*QCR9; BBO-V_III007050), *V. brassicaformis* (*Vb*QCR9; Vbra_943), *Symbiodinium kawagutii* (*Sk*QCR9; Skav217368), *Arabidopsis thaliana* (*At*QCR9; NP_190841), *Homo sapiens* (*Hs*QCR9; NP_037519) and *Saccharomyces cerevisiae* (*Sc*QCR9; NP_011699). Dark shading indicates amino acid identity in ≥70% of the sequences, and light shading indicates amino acid

similarity in ≥70% of the sequences. The positions of predicted transmembrane domains in *Tg*QCR9 (TMPred prediction) and *Sc*QCR9 (TMHMM prediction) are indicated by red boxes.
(TIF)

**S5 Fig. QCR11 alignment.** Alignment of QCR11 homologs from *T. gondii* (*Tg*QCR11; TGME49_214250), *P. falciparum* (*Pf*QCR11; PF3D7_0722700), *P. berghei* (*Pb*QCR11; PBANKA_0620200), *B. bovis* (*Bb*QCR11; BBOV_IV004900), *T. equi* (*Te*QCR11; BEWA_032020), *V. brassicaformis* (*Vb*QCR11; Vbra_12339), *Perkinsum marinus* (*Pm*QCR11; XP_002780203), and *S. kawagutii* (*Sk*QCR11; Skav223196). Dark shading indicates amino acid identity in >70% of the sequences, and light shading indicates amino acid similarity in >70% of the sequences. The position of predicted transmembrane domains in *Tg*QCR11 (TMHMM prediction) is indicated by a red box.
(TIF)

**S6 Fig. QCR12 alignment.** Alignment of QCR12 homologs from *T. gondii* (*Tg*QCR12; TGME49_207170), *P. berghei* (*Pb*QCR12; PBANKA_1341100), *P. falciparum* (*Pf*QCR12; PF3D7_1326000), *T. equi* (*Te*QCR12; BEWA_021660), and *B. bovis* (*Bb*QCR12; BBOV_III005260). Dark shading indicates amino acid identity in ≥80% of the sequences, and light shading indicates amino acid similarity in ≥80% of the sequences. The position of predicted transmembrane domains in *Tg*QCR12 (TMHMM prediction) is indicated by a red box.
(TIF)

**S7 Fig. Generating FLAG tagged *Tg*QCR8, *Tg*QCR9, *Tg*QCR11 and *Tg*QCR12 in *Tg*MPPα-HA strain parasites.** Diagrams depict the 3' replacement strategy to FLAG-tag target genes. sgRNAs were designed to target the *T. gondii* genome near the stop codon of target genes. A plasmid containing the sgRNA and GFP-tagged Cas9 endonuclease was co-transfected into *Tg*MPPα-HA *T. gondii* parasites with a PCR product encoding a FLAG epitope tag flanked by 50 bp of sequence homologous to the regions immediately up- and down-stream of the stop codon. Genomic DNA extracted from wild type (WT) parasites was used as a control in PCRs. **(A)** Forward and reverse primers were used to screen parasite clones for integration of the FLAG tag at the *Tg*QCR8 locus, yielding a 241 bp product in the native locus and a 338 bp product in the modified locus. **(B)** PCR screening using genomic DNA extracted from putative *Tg*QCR8-FLAG parasites (clones 1–6). Clones 1, 4 and 6 yielded PCR products that indicated that these clones had been successfully modified. **(C)** Forward and reverse primers were used to screen parasite clones for integration of the FLAG tag at the *Tg*QCR9 locus, yielding a 288 bp product in the native locus and a 378 bp product in the modified locus. **(D)** PCR screening using genomic DNA extracted from putative *Tg*QCR9-FLAG parasites (clones 1–10). Clone 8 yielded a PCR product that indicated it had been successfully modified. **(E)** Forward and reverse primers were used to screen parasite clones for integration of the FLAG tag at the *Tg*QCR11 locus, yielding a 1.3 kb product in the native locus and a 1.4 kb product in the modified locus. **(F)** PCR screening using genomic DNA extracted from putative *Tg*QCR11-FLAG parasites (clones 1–5). Clone 4 yielded a PCR product that indicated it had been successfully modified. **(G)** Forward and reverse primers were used to screen parasite clones for integration of the FLAG tag at the *Tg*QCR12 locus, yielding a 290 bp product in the native locus and a 398 bp product in the modified locus. **(H)** PCR screening using genomic DNA extracted from putative *Tg*QCR12-FLAG parasites (clones 1–4). Clones 1–3 yielded PCR products that indicated that these clones had been successfully modified.
(TIF)

**S8 Fig. Candidate Complex III subunits are expressed at different levels in *T. gondii* parasites. (A)** Western blot of proteins extracted from *Tg*MPPα-HA/*Tg*QCR11-FLAG, *Tg*MPPα-HA/*Tg*QCR9-FLAG, *Tg*MPPα-HA/*Tg*QCR8-FLAG and *Tg*MPPα-HA/*Tg*QCR12-FLAG parasites, separated by SDS-PAGE, and detected with anti-FLAG and anti-Tom40 (loading control) antibodies. **(B)** Western blot of proteins extracted from *Tg*MPPα-TEV-HA, *Tg*MPPα-HA/*Tg*QCR11-FLAG, *Tg*MPPα-HA/*Tg*QCR9-FLAG, *Tg*MPPα-HA/*Tg*QCR8-FLAG and *Tg*MPPα-HA/*Tg*QCR12-FLAG parasites, separated by BN-PAGE, and detected with anti-FLAG antibodies.
(TIF)

**S9 Fig. Generating an ATc regulated, FLAG-tagged *Tg*QCR11 and r*Tg*QCR11-FLAG/ *Tg*MPPα-HA parasite strains. (A)** Diagram depicting the 3' replacement strategy to FLAG-tag *Tg*QCR11. A sgRNA was designed to target the *T. gondii* genome near the stop codon of *Tg*QCR11. A plasmid containing the sgRNA and GFP-tagged Cas9 endonuclease was co-transfected into *T. gondii* parasites with a PCR product encoding a FLAG epitope tag flanked by 50 bp of sequence homologous to the regions immediately up- and down-stream of the *Tg*QCR11 stop codon. Forward and reverse primers were used to screen parasite clones for integration of the FLAG tag at the *Tg*QCR11 locus, yielding a 1.3 kb product in the native locus and a 1.4 kb product in the modified locus. **(B)** PCR screening using genomic DNA extracted from putative *Tg*QCR11-FLAG parasites (clones 1–12). Clone 6 yielded a PCR product that indicated it had been successfully modified. Genomic DNA extracted from wild type (WT) parasites was used as a control. **(C)** Diagram depicting the promoter replacement strategy to generate ATc-regulated *Tg*QCR11. A sgRNA was designed to target the *T. gondii* genome near the start codon of *Tg*QCR11. A plasmid containing the sgRNA and GFP-tagged Cas9 endonuclease was co-transfected into *T. gondii* parasites with a PCR product encoding the ATc regulated 't7s4' promoter, which contains 7 copies of the Tet operon and a Sag4 minimal promoter, flanked by 50 bp of sequence homologous to the regions immediately up- and down-stream of the *Tg*QCR11 start codon. The PCR product also contain a 'spacer' region that separates the regulatable promoter from the native promoter of the *Tg*QCR11 gene to enable sufficient regulation. Forward and reverse primers were used to screen parasite clones for successful integration of the regulatable promoter at the *Tg*QCR11 locus, yielding a 1.4 kb product in the native, FLAG-tagged locus and a 3.3 kb product in the modified locus. **(D)** PCR screening using genomic DNA extracted from putative r*Tg*QCR11-FLAG parasites (clones 1–9). Clones 1, 2, 4–7 yielded PCR products that indicated that these clones had been successfully modified. Genomic DNA extracted from wild type (WT) parasites was used as a control. **(E)** Diagram depicting the 3' replacement strategy to HA-tag *Tg*MPPα. A sgRNA was designed to target the *T. gondii* genome near the stop codon of *Tg*MPPα. A plasmid containing the sgRNA and GFP-tagged Cas9 endonuclease was co-transfected into r*Tg*QCR11-FLAG *T. gondii* parasites with a PCR product encoding a HA epitope tag flanked by 50 bp of sequence homologous to the regions immediately up- and down-stream of the *Tg*MPPα stop codon. Forward and reverse primers were used to screen parasite clones for integration of the HA tag at the *Tg*MPPα locus, yielding a 274 bp product in the native locus and a 387 bp product in the modified locus. **(F)** PCR screening using genomic DNA extracted from putative r*Tg*QCR11/*Tg*MPPα-HA parasites (clones 1–3). Clones 2 and 3 yielded PCR products that indicated that these clones had been successfully modified. Genomic DNA extracted from wild type (WT) parasites was used as a control.
(TIF)

**S10 Fig. Defects in mOCR observed upon knockdown of *Tg*QCR11 are not caused by general defects in mitochondrial morphology or parasite metabolism. (A)** Immunofluorescence assays assessing mitochondrial morphology in r*Tg*QCR11-FLAG/*Tg*MPPα-HA

parasites grown in the absence of ATc (top) or in the presence of ATc for 3 days (bottom). The outer mitochondrial membrane was labelled using antibodies against *Tg*Tom40 (red), and the inner mitochondrial membrane was labelled using anti-HA antibodies to detect *Tg*MPPα-HA. Images are representative of 100 four-cell vacuoles examined in 2 independent experiments; scale bar represents 2 μm. **(B)** Basal extracellular acidification rate (ECAR) of WT parasites grown in the absence of ATc or in the presence of ATc for 3 days (blue), and r*Tg*QCR11-FLAG/*Tg*MPPα-HA parasites grown in the absence of ATc or in the presence of ATc for 1–3 days (orange). A linear mixed-effects model was fitted to the data and values depict the least squares mean ± 95% CI of three independent experiments. ANOVA followed by Tukey's multiple pairwise comparisons test was performed, with relevant *p* values shown. (TIF)

**S11 Fig. Generating TEV-HA tagged *Tg*QCR8 and *Tg*QCR12 in r*Tg*QCR11-FLAG parasites.** Diagrams depict the 3' replacement strategy to TEV-HA-tag target genes. sgRNAs were designed to target the *T. gondii* genome near the stop codon of target genes. A plasmid containing the sgRNA and GFP-tagged Cas9 endonuclease was co-transfected into r*Tg*QCR11-FLAG *T. gondii* parasites with a PCR product encoding a TEV-HA epitope tag flanked by 50 bp of sequence homologous to the regions immediately up- and down-stream of the stop codon. Genomic DNA extracted from wild type (WT) parasites was used as a control in PCRs. **(A)** Forward and reverse primers were used to screen parasite clones for integration of the TEV-HA tag at the *Tg*QCR8 locus, yielding a 241 bp product in the native locus and a 377 bp product in the modified locus. **(B)** PCR screening using genomic DNA extracted from putative r*Tg*QCR11-FLAG/*Tg*QCR8-TEV-HA parasites (clones 1–12). Clone 9 yielded PCR products that indicated it had been successfully modified. **(C)** Forward and reverse primers were used to screen parasite clones for integration of the TEV-HA tag at the *Tg*QCR12 locus, yielding a 290 bp product in the native locus and a 425 bp product in the modified locus. **(D)** PCR screening using genomic DNA extracted from putative r*Tg*QCR11-FLAG/*Tg*QCR12-TEV-HA parasites (clones 1–6). Clones 1–2 and 4–6 yielded PCR products that indicated that they had been successfully modified. (TIF)

**S12 Fig. The catalytic Complex III subunit *Tg*CytC1 interacts with *Tg*QCR11.** Western blots of proteins extracted from r*Tg*QCR11-FLAG/*Tg*CytC1-HA parasites, and subjected to immunoprecipitation using anti-HA (anti-HA IP) or anti-FLAG (anti-FLAG IP) antibody-coupled beads. Extracts include samples before immunoprecipitation (Total), samples that did not bind to the anti-HA or anti-FLAG beads (Unbound), and samples that bound to the anti-HA or anti-FLAG beads (Bound). Samples were separated by SDS-PAGE, and probed with anti-HA antibodies to detect *Tg*CytC1-HA, anti-FLAG to detect *Tg*QCR11, and anti-*Tg*Tom40 as a control to detect an unrelated mitochondrial protein. (TIF)

**S13 Fig. *Tg*QCR11 is important for Complex III integrity. (A-D)** Quantification of BN-PAGE western blots from Fig 6. **(A)** r*Tg*QCR11-FLAG/*Tg*MPPα-HA, **(B)** r*Tg*QCR11-FLAG/*Tg*QCR12-TEV-HA, **(C)** r*Tg*QCR11-FLAG/*Tg*QCR8-TEV-HA and **(D)** r*Tg*QCR11-FLAG/*Tg*CytC1-HA parasites were grown in the absence of ATc or in the presence of ATc for 1–3 days. Band intensities were normalized to the matched SDS-PAGE *Tg*Tom40 control and expressed as a percent of the day zero control. Columns represent the mean ± SD of at least 2 independent experiments, with individual values depicted. **(E-H)** Quantification of SDS-PAGE western blots from Fig 6. **(E)** r*Tg*QCR11-FLAG/*Tg*MPPα-HA, **(F)** r*Tg*QCR11-FLAG/*Tg*QCR12-TEV-HA, **(G)** r*Tg*QCR11-FLAG/*Tg*QCR8-TEV-HA and **(H)**

r*Tg*QCR11-FLAG/*Tg*CytC1-HA parasites were grown in the absence of ATc or in the presence of ATc for 1–3 days. Band intensities were normalized relative to the *Tg*Tom40 control and expressed as a percent of the day zero control. Columns represent the mean ± SD of at least 2 independent experiments, with individual values depicted. r*Tg*QCR11-FLAG is shown in gray tones and HA-tagged proteins are shown in red tones.
(TIF)

**S1 Table. Extended data from the mass spectrometry-based proteomic analysis of the *Tg*MPPα complex.** Tab 1: List of proteins identified in the mass spectrometry-based proteomic analysis of proteins purified from the *Tg*MPPα-TEV-HA and *Tg*Cox2a-TEV-HA immunoprecipitations. Included are the ToxoDB (www.toxodb.org) accession number, the name assigned to the protein in this manuscript if applicable, the total precursor intensity (area) for each biological replicate, the average total precursor intensity (area) of the three replicates, the fold change (average *Tg*MPPα/average *Tg*Cox2a), and the predicted mass of the identified protein. Proteins are highlighted based on whether they were highly enriched in *Tg*MPPα-TEV-HA (orange) or *Tg*Cox2a-TEV-HA (blue) pulldown. Tab 2: A list of the log fold change (logFC) and *p* values calculated for each protein identified in all replicates of the *Tg*MPPα-TEV-HA and *Tg*Cox2a-TEV-HA immunoprecipitations following EdgeR analysis.
(XLSX)

**S2 Table. Sequences of oligonucleotides (primers) and gBlocks used in this study.**
(PDF)

## Acknowledgments

We thank Harpreet Vohra and Michael Devoy (ANU) for assistance with flow cytometry, Michael Devoy for assistance with establishing the XFe96 assays, Teresa Neeman from the ANU Statistical Consulting Unit for assistance with data analysis, Chris Tonkin (WEHI) and Yi Xue (ANU) for sharing vectors, the ANU Toxo lab for comments on the manuscript, and the 2019 ANU Cell Biology course for contributing to the generation of parasite strains. We are grateful to EuPathDB for providing numerous datasets and search tools.

## Author Contributions

**Conceptualization:** Jenni A. Hayward, Esther Rajendran, Giel G. van Dooren.

**Data curation:** Jenni A. Hayward, Pierre Faou.

**Formal analysis:** Jenni A. Hayward, Esther Rajendran, Soraya M. Zwahlen, Giel G. van Dooren.

**Funding acquisition:** Jenni A. Hayward, Esther Rajendran, Giel G. van Dooren.

**Investigation:** Jenni A. Hayward, Esther Rajendran, Soraya M. Zwahlen, Pierre Faou, Giel G. van Dooren.

**Methodology:** Jenni A. Hayward, Esther Rajendran, Pierre Faou, Giel G. van Dooren.

**Project administration:** Jenni A. Hayward, Esther Rajendran, Giel G. van Dooren.

**Resources:** Pierre Faou, Giel G. van Dooren.

**Supervision:** Esther Rajendran, Giel G. van Dooren.

**Validation:** Jenni A. Hayward, Esther Rajendran, Soraya M. Zwahlen, Giel G. van Dooren.

**Visualization:** Jenni A. Hayward, Soraya M. Zwahlen, Giel G. van Dooren.

**Writing – original draft:** Jenni A. Hayward.

**Writing – review & editing:** Jenni A. Hayward, Soraya M. Zwahlen, Pierre Faou, Giel G. van Dooren.

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
