## [Decision Letter · Decision Letter 0]

27 Jun 2020

Dear Dr. van Dooren,

Thank you very much for submitting your manuscript "Divergent features of the coenzyme Q:cytochrome c oxidoreductase complex in Toxoplasma gondii parasites." for consideration at PLOS Pathogens. As with all papers reviewed by the journal, your manuscript was reviewed by members of the editorial board and by several independent reviewers. In light of the reviews (below this email), we would like to invite the resubmission of a significantly-revised version that takes into account the reviewers' comments.

The reviewers are enthusiastic about the work both from a fundamental point of view as well as for its potential to define novel targets for therapeutic intervention. However, they identified some issues that will have to be addressed prior to consideration for publication. These include the need to discuss further some of your findings in the broader context of the literature (Rev #2 points 2 and 3). In addition, at least two main points will need to be address experimentally. Cyt c1 should be included as bait for IP/MS studies to consolidate and further validate the composition of the complex III (Rev #1 point 1). The possible existence of super-complexes should be explored under variable detergent concentration and if they exist assess the influence of QCR10 and QCR11 (Rev#2 point 1)

We cannot make any decision about publication until we have seen the revised manuscript and your response to the reviewers' comments. Your revised manuscript is also likely to be sent to reviewers for further evaluation.

Sincerely,

Dominique Soldati-Favre

Associate Editor

PLOS Pathogens

Vern Carruthers

Section Editor

PLOS Pathogens

Kasturi Haldar

Editor-in-Chief

PLOS Pathogens

orcid.org/0000-0001-5065-158X

Michael Malim

Editor-in-Chief

PLOS Pathogens

orcid.org/0000-0002-7699-2064

Reviewer's Responses to Questions

**Part I - Summary**

Reviewer #1: This manuscript presents a proteomic and functional analysis of interacting mitochondrial proteins in Toxoplasma gondii parasites that form a 675 kDa complex and appear to be components of respiratory complex III. The authors identify core, canonical members of complex III that appear to interact with novel Apicomplexa-specific protein components. One of these novel components, termed ApiQCR10, is essential for mitochondrial respiration and stability of the 675 kDa complex.

The mitochondrial electron transport chain (ETC) is a major drug target for treating T. gondii and P. falciparum. However, the composition and molecular features of ETC complexes have not been well-studied despite intense scrutiny in yeast and mammals. Defining novel features of ETC complexes in Apicomplexan parasites will uncover evolutionary differences that might be exploited as new parasite-specific drug targets. The present manuscript is an important step forward in defining novel parasite-specific components of ETC complex III. Although I am enthusiastic overall about the novel insights provided by this work, I have several questions and critiques regarding the approach and conclusions in this paper.

Reviewer #2: The manuscript by Hayward et al. provides proteomic and genetic evidence showing divergent subunit composition of the Complex III (CIII) of Toxoplasma mitochondrial electron transport chain (ETC). Inasmuch as CIII is the target of anti-parasitic drugs for apicomplexan parasites, this study is of interest. The study is well-planned and is carried out with rigor. CIII is one of the most extensively investigated components of ETC with its structure from several organisms resolved, its catalytic cycle understood in exquisite detail, and mechanisms of its inhibition by a plethora of inhibitors established at the molecular level. At this point, no structures of CIII from any apicomplexan parasites have been established, so having additional information on its subunit composition is useful. The manuscript, however, ignores certain well-known roles of unique supernumerary subunits of respiratory complexes from different organisms, and in process, oversells the significance of having found subunits that appear to limited to myxozoan lineage. Addressing the points raised below would significantly improve the manuscript and contextualize it in reference to the vast amount of information available on ETC in general and CIII in particular.

Reviewer #3: In this manuscript, Hayward et al. use a variety of molecular, biochemical and cellular approaches to assess the composition of the mitochondrial coenzyme Q: cytochrome c oxdoireductase complex (complex III) in Toxoplasma gondii. Interestingly, complex III is a major drug target of this parasite yet the enzyme has not been well studied. Thus, new information into the composition of the complex and its differences to mammalian host complex III can provide important fundamental and applied outcomes.

Initially, the authors utilize clever protein tagging plus co-immunoprecipitation techniques coupled with blue-native-PAGE and proteomic analyses to identify novel complex III assembly components in T. gondii. The authors identified 9 (strongly/partly) conserved complex III subunits along with two novel proteins specific to apicomplexans. The latter, TgApiQCR10 and TgApiQCR11, were identified through neat bioinformatic analyses and validated by fluorescence microscopy, co-immunoprecipitation and blue-native-PAGE analyses. TgApiQCR10 was subsequently investigated and found to be critical for T. gondii proliferation and mitochondrial respiration due to its role in complex III assembly.

The manuscript is well written and the data is very clear. I have only a few minor concerns/comments that the authors may wish address.

**Part II – Major Issues: Key Experiments Required for Acceptance**

Reviewer #1: 1. MPP has been identified as a stable member of complex III in other organisms, but its interactions with complex III have not previously been studied in Apicomplexa. It seems odd then that the authors chose MPP as bait for identifying the protein components of complex III, rather than one of the main complex III components such as cyt c1, especially as the authors recently published a tagged cyt c1 line (ref. 25). In this regard, inclusion of cyt c1 as bait for IP/MS studies and as positive control for reciprocal IP studies in Fig. 3 would be a strong complement to the MPP data to triangulate complex III composition and tighten the conclusion that the identified 675 kDa complex indeed reflects complex III.

2. Is complex IV the optimal negative control to use for IP/MS experiments? Super-complex formation between complexes III and IV occurs in other organisms. Although unstudied in Apicomplexa, any stable interactions between these complexes in T. gondii could lead to false negatives in the proteomic analysis of complex III components. Tom40 was used as a negative control in Fig. 3- why not use it as the negative control for IP/MS experiments? Conversely, why wasn’t complex IV used as negative control in Fig. 3?

3. Fig. 3A lacks an appropriate negative control. The authors should ideally show an anti-FLAG blot of a BN-PAGE with the parental MPP-HA only line to rule out antibody cross-reactivity.

Reviewer #2: 1. Investigations on ETC complexes from multiple organisms have shown the complexes assemble into very large super-complexes, sometimes referred to respirasomes. Cryo-EM structures of some of them have been solved (see e.g. PMID:27654917, PMID:27912063 and PMID:30598554). Supernumerary subunits seem to be critical for the assembly of these super-complexes. Authors here did not detect super-complexes in blue native (BN) gel electrophoresis, which could likely be due to the solubilization conditions they used; respirasomes are sensitive to detergent concentrations used. Clearly, proteomic analyses of the complexes require harsher detergent solubilization to avoid artefactual associations, but this could also eliminate associations that are significant. While it is possible that Toxoplasma ETC may not form super-complexes, this would be unusual. Authors should be encouraged to carry out BN gel electrophoresis under low to high detergent concentrations to assess the presence or absence of super-complexes. In addition, effects of knockdown of QCR10 and QCR11 on potential super-complexes should be assessed.

2. Unlike the mammalian ETC, apicomplexans (just like the budding yeasts) lack Complex I but have a single subunit NADH dehydrogenase. This leads to a different arrangement of supercomplexes in the yeast. Proteins Rcf1 and Rcf2 are critical for the formation of CIII and CIV supercomplex in the yeast (PMID:22310663), and Rcf2 was recently shown to be part of the supercomplex by cryo-EM (PMID:32291341). These 2 yeast proteins are unique and not seen in other organisms. Authors should interpret their findings of unique QCR proteins in myxozoans in light of these and other results showing the role of such supernumerary subunits in ETC assembly and functions.

3. Authors advance their findings to suggest that they could form the basis for antiparasitic drug discovery. Selective toxicity of anti-parasitic drugs against CIII arises from the structural differences between the Qo and Qi sites within the cytochrome b encoded by the parasite and their hot mitochondrial genome, something that was noted 27 years ago (PMID:8459834). Of the numerous CIII inhibitors known, essentially all work through inhibition of either Qo or Qi site of the enzyme. It would be important for the authors to tame their argument and present it in the context of the vast literature on CIII inhibitors and their mode of action.

Reviewer #3: (No Response)

**Part III – Minor Issues: Editorial and Data Presentation Modifications**

Reviewer #1: 4. A Scheme summarizing canonical subunits and membrane topology of complex III would be helpful in the introduction, especially to set the stage for what compoments may be conserved or divergent in Toxoplasma and also to suggest possible roles for the novel subunits identified herein.

5. It is confusing to state that TGGT1_214250 lacks sequence homology to known proteins but refer to it as “TgApiQCR10”. QCR10 is a known complex III sub-unit in yeast (https://www.yeastgenome.org/locus/S000003529).

6. Lines 160-161: The HHPRED analysis probes sequence not structural similarity. No structures have been determined for the T. gondii proteins so it is impossible to evaluate structural similarity. Rather, the HMM analysis at HHPRED identifies remote sequence similarity to proteins whose structures have been deposited at the PDB as a basis for generating a homology model.

7. The authors should ideally give the e-value and/or probability scores returned by HHPRED for sequence similarity for TGGT1_227910 to yeast QCR8 and TGGT1_201880 to yeast QCR9.

8. Fig. 6D-F: I suggest that the authors quantify band intensity normalized to Tom40 to provide a more quantitative basis for conclusions about the effect of ApiQCR10 KD on stability of QCR8, QCR11, and MPP. The current qualitative comparisons are difficult to evaluate, especially in Fig. 6F where ApiQCR10 KD appears weak.

9. For the experiment in Fig. S9B, how does the viability/recovery of parasites after ApiQCR10 KD compare to 24-48-hr treatment with atovaquone, which should phenocopy loss of complex III activity for on-target activity? Is viability of the KD parasites due to residual protein expression and trace complex III function or an ability to switch to glycolysis? Related, what does parasite viability despite QCR10 KD and loss of mitochondrial respiration suggest or imply about the efficacy of atovaquone or novel treatments that target ETC function?

10. It would be helpful if the authors can give the acrylamide % used for BN-PAGE experiments.

Reviewer #2: 1. Line 55: Clarify that electrons are passed from *reduced* CoQ to the *heme* within cytochrome b protein.

2. Lines 61 and 66: there are better references for these statements than #9 and #10.

3. Lines 75 and 76: Discovery of novel proteins in CIII does not necessarily open avenues for drug development. No examples are know for such inhibitors working on CIII.

4. Line 139: The organelle in Cryptococcus parvum is termed a mitoplast and is not considered an authentic mitochondrion, reduced or otherwise.

5. QCR10 is appears to be expressed at a much greater level compared to other subunits, yet its signal in the BN gel appears to be stoichiometric. This could suggest additional functions associated with this protein. Authors should acknowledge this possibility.

6. Authors should clearly state in the Results sections that OCR and ECAR measurements were done in isolated tachyzoites.

Reviewer #3: 1. TgApiQCR11 levels are a lot lower than other complex III subunits (Sfig 7). The authors should consider that this protein is an assembly factor rather than a stoichiometric subunit. I understand that the protein appears to assemble into a complex that is similar in size to complex III, but since the lane in Fig. 3A is independent of the other samples, this cannot be clarified. It would therefore be appropriate to repeat the BN-PAGE blot with all samples on the same blot and show different exposures to capture the high molecular weight complex signal.

2. The authors could also validate/assess whether complex III exists in a supercomplex as found in other organisms by using digitonin instead of DDM. The migration of the subunits with that larger form (especially TgApiQCR11) would help clarify that it is a bona fide complex III subunit.

PLOS authors have the option to publish the peer review history of their article (what does this mean?). If published, this will include your full peer review and any attached files.

Reviewer #1: No

Reviewer #2: No

Reviewer #3: No
---

## [Decision Letter · Decision Letter 1]

3 Dec 2020

Dear Dr. van Dooren,

We are pleased to inform you that your manuscript 'Divergent features of the coenzyme Q:cytochrome c oxidoreductase complex in Toxoplasma gondii parasites.' has been provisionally accepted for publication in PLOS Pathogens.

Best regards,

Dominique Soldati-Favre

Associate Editor

PLOS Pathogens

Vern Carruthers

Section Editor

PLOS Pathogens

Kasturi Haldar

Editor-in-Chief

PLOS Pathogens

orcid.org/0000-0001-5065-158X

Michael Malim

Editor-in-Chief

PLOS Pathogens

orcid.org/0000-0002-7699-2064

Reviewer Comments (if any, and for reference):

Reviewer's Responses to Questions

**Part I - Summary**

Reviewer #1: The authors have carefully responded to the critiques and suggestions in the prior review, including addition of new experiments (especially addition of the requested cyt c1 IP experiment), analyses, and discussions. This study was executed well and provides a much-needed inventory of conserved and novel subunits of complex III in T. gondii and the functional consequences upon knock-down of at least one of the Apicomplexan-specific subunits. Knowledge of subunit composition for complex III will provide a powerful and impactful basis for exploring new strategies to target complex III in different pathogens.

Reviewer #2: Authors have adequately addressed points raised by this reviewer.

**Part II – Major Issues: Key Experiments Required for Acceptance**

Reviewer #1: The authors have addressed all major issues in the prior review.

Reviewer #2: (No Response)

**Part III – Minor Issues: Editorial and Data Presentation Modifications**

Reviewer #1: The authors have addressed all minor issues in the prior review.

Reviewer #2: (No Response)

PLOS authors have the option to publish the peer review history of their article (what does this mean?). If published, this will include your full peer review and any attached files.

Reviewer #1: No

Reviewer #2: No

---

## [Editor Report · Acceptance letter]

27 Jan 2021

Dear Dr. van Dooren,

We are delighted to inform you that your manuscript, "Divergent features of the coenzyme Q:cytochrome c oxidoreductase complex in Toxoplasma gondii parasites.," has been formally accepted for publication in PLOS Pathogens.

Best regards,

Kasturi Haldar

Editor-in-Chief

PLOS Pathogens

orcid.org/0000-0001-5065-158X

Michael Malim

Editor-in-Chief

PLOS Pathogens

orcid.org/0000-0002-7699-2064